# Optimal Decision Tree Pruning Revisited: Algorithms and Complexity

**Juha Harviainen** [* 1]  **Frank Sommer** [* 2]  **Manuel Sorge** [* 2]  **Stefan Szeider** [* 2]

## Abstract

We present a comprehensive classical and parameterized complexity analysis of decision tree pruning operations, extending recent research on the complexity of learning small decision trees. Thereby, we offer new insights into the computational challenges of decision tree simplification, a crucial aspect of developing interpretable and efficient machine learning models. We focus on fundamental pruning operations of subtree replacement and raising, which are used in heuristics. Surprisingly, while optimal pruning can be performed in polynomial time for subtree replacement, the problem is NP-complete for subtree raising. Therefore, we identify parameters and combinations thereof that lead to fixed-parameter tractability or hardness, establishing a precise borderline between these complexity classes. For example, while subtree raising is hard for small domain size $D$ or number $d$ of features, it can be solved in $D^{2d} \cdot |I|^{\mathcal{O}(1)}$ time, where $|I|$ is the input size. We complement our theoretical findings with preliminary experimental results, demonstrating the practical implications of our analysis.

## 1. Introduction

Decision trees are fundamental data structures used to describe, classify, and generalize data (Larose, 2014; Murthy, 1998; Quinlan, 1986). They are widely used in machine learning due to their interpretability and efficiency (Breiman et al., 1984). Towards explainable AI, one prefers small decision trees over large ones as they provide more concise and understandable models (Rudin, 2019;

---

[*]Equal contribution  [1]Department of Computer Science, University of Helsinki, Helsinki, Finland  [2]Institute of Logic and Computation, TU Wien, Austria. Correspondence to: Juha Harviainen <juha.harviainen@helsinki.fi>, Frank Sommer <fsommer@ac.tuwien.ac.at>, Manuel Sorge <manuel.sorge@ac.tuwien.ac.at>, Stefan Szeider <sz@ac.tuwien.ac.at>.

*Proceedings of the 42nd International Conference on Machine Learning*, Vancouver, Canada. PMLR 267, 2025. Copyright 2025 by the author(s).

Holzinger et al., 2020). Recent advancements in algorithms have made it feasible to compute decision trees that are optimal with respect to various optimization goals (e.g., Narodytska et al., 2018; Demirovic et al., 2022; McTavish et al., 2022). With that, the algorithmics and parameterized complexity of the underlying optimization problems were studied intensively (Staus et al., 2025; Ordyniak & Szeider, 2021; Kobourov et al., 2025; Eiben et al., 2023; Komusiewicz et al., 2023a; Ordyniak et al., 2024; Gahlawat & Zehavi, 2024; Komusiewicz et al., 2025), feeding back into practical advances (Staus et al., 2025).

Large datasets still require heuristic optimization techniques, however. Commonly used heuristics to compute decision trees for given data recursively split the input data based on certain criteria such as reduction in entropy (Quinlan, 1986; Breiman et al., 1984; Mingers, 1989). The resulting large trees often overfit, and so the heuristics then prune them, that is, they delete nodes to decrease the size while maintaining good classification performance. In other words, they heuristically solve optimization problems in which they balance some form of the two goals of maximizing the number of pruned nodes and minimizing the number of introduced errors. This motivates studying these optimization problems themselves.

While the algorithmics of computing optimal decision trees from scratch is reasonably well understood, the algorithmics of optimally pruning a given decision tree has received scant attention. Our goal is to initiate a rigorous algorithmic study of the latter, highlighting what properties make the underlying problems hard or tractable, and pointing to promising algorithmic approaches that may be developed further into practical implementations.

There are two main operations that heuristics apply, subtree replacement and subtree raising (see below for details), and we study optimally pruning trees under each of these. An overview of our results is as follows. It was already known that optimally replacing subtrees is polynomial-time solvable (Almuallim, 1996), so we study the running time in more detail and give an improved algorithm that is linear instead of quadratic in the tree size if the number $k$ of pruned nodes or number $t$ of misclassifications is small. As a side result, we give a quicker algorithm for classifying examples with a given tree based on heavy–light decom-

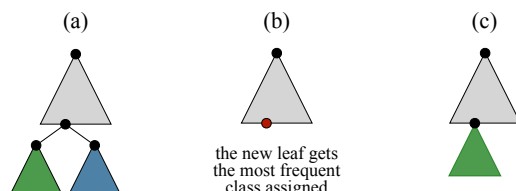

*Figure 1.* Illustration of the two pruning operations. (a) shows the input tree $T$. (b) shows the result of one subtree replacement operation. (c) shows the result of one subtree raising operation.

positions (Sleator & Tarjan, 1983). In contrast, we show that optimally pruning a decision tree with subtree raising is NP-complete. In general we thus cannot expect efficient algorithms and therefore investigate which aspects of the problem make it hard or tractable. In this regard, we completely classify the influence of natural single parameters, such as $k$, $t$, the number $d$ of features, or the domain size $D$ on the complexity of raising subtrees optimally. Further, we almost completely classify all pairs and triples of parameters. For instance, we show that we cannot expect efficient algorithms for optimally raising subtrees if $D$ or $d$ is small, however, there is a prospect for an efficient algorithm if they are both small at the same time. This latter algorithm might be relevant for practice: We provide a proof-of-concept implementation and use it on standard benchmark data to show that heuristics achieve an almost optimal tradeoff between the number of pruned nodes and introduced classification errors.

Our results offer new insights into the computational challenges of decision tree simplification and provide a theoretical foundation for developing more efficient pruning algorithms. Our results contribute to the growing body of work on the theoretical foundations of interpretable ML, which is crucial for developing trustworthy AI systems.[1]

**Problem statement.** We study two types of pruning operations that are used by implementations in well-established machine-learning libraries (see Figure 1 for illustrations), focusing on decision trees that are binary trees throughout the paper. Let $T$ be a decision tree for a set $E \subseteq \mathbb{R}^d$ of examples labeled via $\lambda \colon E \to \{\mathsf{blue}, \mathsf{red}\}$ by two[2] classes $\mathsf{blue}$ and $\mathsf{red}$ and let $w \in V(T)$ an inner (non-leaf) node of $T$. Here, for a graph $G$, by $V(G)$ we refer to its vertex set.

A subtree *replacement operation* applied to $w$ removes $w$ and its entire subtree from $T$ and replaces it by a new leaf $u$ which has the most frequent class label of all examples in the subtree of $w$, that is, $u$ receives color $\mathsf{blue}$ if the

set $E[T, w]$ of examples classified in the subtree rooted at $w$ contains at least as many $\mathsf{blue}$ examples as $\mathsf{red}$ examples, and otherwise $u$ receives color $\mathsf{red}$. Replacement is a basic pruning operation and used in CART (Breiman et al., 1984) and C4.5 (Quinlan, 1993), for example.

A subtree *raising operation* applied to $w$ removes $w$ and its entire left or right subtree from $T$. In other words, we choose a child $u$ of $w$ and then we remove the subtree rooted at $w$ and replace it by the subtree rooted at $u$. Subtree raising is implemented in the well-known decision tree heuristics C4.5 (Quinlan, 1993), C5.0, and J48 (Witten et al., 2011).

We now formulate the optimization problems implicitly solved by the tree-pruning heuristics as search problems: We aim to prune $k$ inner nodes while satisfying an upper bound $t$ on the number of resulting classification errors.[3] Algorithms solving these problems can also perform error minimization.

DECISION TREE REPLACEMENT (DTREP)
*Instance:* A training data set $(E, \lambda)$, a decision tree $T$ for $(E, \lambda)$, and $k, t \in \mathbb{N}$.
*Question:* Can we perform replacement operations that prune exactly $k$ inner nodes such that the resulting tree $T'$ has at most $t$ errors?

DECISION TREE RAISING (DTRAIS$_=$)
*Instance:* A training data set $(E, \lambda)$, a decision tree $T$ for $(E, \lambda)$, and $k, t \in \mathbb{N}$.
*Question:* Can we perform raising operations that prune exactly $k$ inner nodes such that the resulting tree $T'$ has at most $t$ errors?

For technical reasons (see the preliminaries), replacing *at least* $k$ nodes has the same complexity as replacing *exactly* $k$. This is not so for raising, and thus we also study the variant DTRAIS$_\geq$ where we perform at least $k$ raising operations.

**Results for replacement.** It is known that DTREP can be solved in $\mathcal{O}\big((n + \ell)s\big)$ time (Almuallim, 1996), where $n$ is the number of input examples, $s$ the size of the input tree and $\ell = s - k$ the size of the tree after pruning. This means that the running time is quadratic in the tree size. We show that one can achieve time linear in the size, that is, $\mathcal{O}\big((n + \min\{k^2, t^2\}) \cdot s\big)$ time, if $k$ or $t$ is small (Theorem 3.1). As a side result, we show that classifying a given example can be done in time $\mathcal{O}(d \log^2 s)$ after $\mathcal{O}(ds)$-time preprocessing, where $d$ is the number of

---

[1]A continuously updated version of our paper is available on arXiv (Harviainen et al., 2025a) and the related source code for replicating the experiments on Zenodo (Harviainen et al., 2025b).

[2]In general our algorithms can also be used in the multiclass setting; see Section 7 for a short discussion.

[3]Throughout, we use the following intuitive equivalence between the number of operations and the number of pruned nodes: A replacement operation that removes $k$ inner nodes can be simulated by $k$ replacement operations applied to inner nodes that have 2 leaves as children. Similarly, a raising operation that removes $k$ inner nodes can be simulated by $k$ raising operations applied to inner nodes that have 2 children and at least one of them is a leaf.

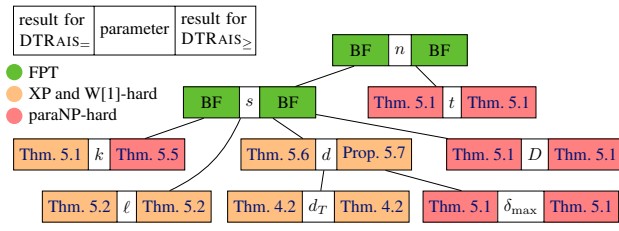

*Figure 2.* A Hasse diagram of the single parameter relations and results for DTRAIS$_=$, DTRAIS$_\geq$: A parameter $p$ has an edge to a lower parameter $q$ if there is a function $f$ such that after straightforward preprocessing we have $q \leq f(p)$. The corresponding theorems and propositions are given in the boxes; for hardness the reference is in the highest box for which hardness holds, for (FPT or XP) tractability the reference is in the lowest box for which tractability holds. BF is for brute-force algorithm.

features (Lemma 3.2). This improves on the straightforward $\mathcal{O}(s)$-time algorithm. Given these polynomial-time results, it is interesting to extend them to replacing subtrees in decision-tree ensembles, which have received a tremendous amount of attention for their simplicity and improved accuracy over plain decision trees (Breiman, 2001; Rokach, 2016). However, we show that efficiently pruning ensembles is unlikely, since the problem is NP-complete even if they contain only two trees (Theorem 3.4).

**Results for raising.** In contrast to the tractability of DTREP, surprisingly DTRAIS$_=$ and DTRAIS$_\geq$ turned out to be NP-complete. Hence, we studied the parameterized complexity, determining the influence of the most natural parameters on the problems' complexity (Gottlob et al., 2002; Flum & Grohe, 2006; Niedermeier, 2006; Cygan et al., 2015; Downey & Fellows, 2013). There are three main levels of influence that a parameter $p$ can have when a problem is NP-hard: ideally (1) fixed-parameter tractability (FPT), that is, there is an algorithm with $f(p) \cdot |I|^{O(1)}$ running time, or (2) W[1]-hardness and XP-tractability, that is, there is an algorithm with running time $f(p) \cdot |I|^{f(p)}$ and it is likely not possible to remove the dependence of the exponent on $p$, and (3) paraNP-hardness, that is, even for constant values of $p$ the problem is NP-hard.

Natural parameters for this analysis are the size $s$ of the initial unpruned tree $T$, the lower bound $k$ on the removed inner nodes, and the upper bound $t$ of errors of the pruned tree. A dual parameter to $k$ is the upper bound $\ell$ on the size of the tree after pruning ($k + \ell = s$). Further natural parameters are a priori related to the input dataset, but actually we may assume that these parameters refer only to the input trees, not to the data itself, see Section 2: the number $d$ of features, the number $n$ of examples, and the maximum domain size $D$, that is, the maximum number of different values that a feature can attain. Furthermore, we consider

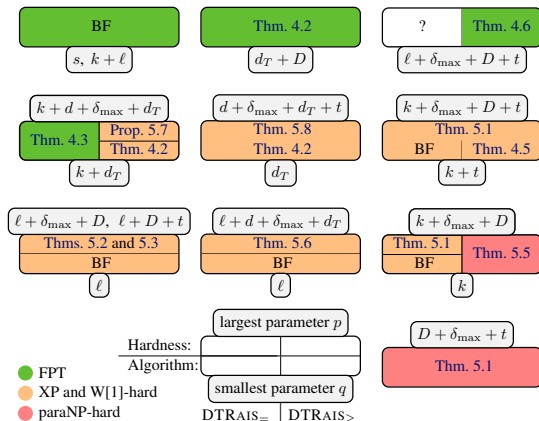

*Figure 3.* Overview of our results for DTRAIS$_=$, DTRAIS$_\geq$. For each box $q$ is the smallest parameter required to achieve an FPT or XP algorithm, and $p$ is the largest parameter such that W[1]-hardness or paraNP-hardness holds. Also, each parameter combination which is not smaller than $p$ and not larger than $q$ leads to the same classification result. Consequently, for parameters $q$ leading to an FPT-algorithm, all parameters which are not smaller than $q$ also lead to an FPT-algorithm. BF is for brute-force algorithm.

the parameter $\delta_{\max}$, the maximum number of features in which two examples of different classes differ.[4] Also, we consider the largest number $d_T$ of different features that occur on a root-to-leaf path in the input tree.

Figure 2 shows an overview over the relations between all parameters together with our complexity results for DTRAIS$_=$ and DTRAIS$_\geq$ for individual parameters; in fact, we completely classify the two problems with respect to the three levels of influence that the parameters can have. Apart from two trivial tractability results for $n$ and $s$, assuming all other individual parameters to be small still yields intractable problems. Notably, DTRAIS$_\geq$ remains NP-hard, even for pruning at least $k = 0$ nodes. Given such broadly negative results, we also consider combinations of two or more parameters, see Figure 3 for an overview. Indeed, we obtain an almost full classification for pairs and triples of parameters. Among several tractability results, we obtain an algorithm with $D^{2d_T} \cdot |I|^{\mathcal{O}(1)}$ running time.

This latter algorithm is particularly interesting in combination with measurements that show that the parameters $D$ and $d_T$ are small in benchmark data for computing optimal decision trees. Thus we provide a proof-of-concept implementation and use it to compute the complete Pareto-front of the optimal tradeoffs between the number $k$ of pruned nodes and number $t$ of classification errors. This allows us for the first time to measure the quality of the heuristic pruning techniques, showing that they achieve almost optimal

---

[4]See Ordyniak & Szeider (2021, Table 1) and Staus et al. (2025, Table 3) for indication that this parameter is small in practical data.

tradeoffs in our data.

## 2. Preliminaries

For $m \in \mathbb{N}$ we write $[m] := \{1, 2, \ldots, m\}$ and $[0, m] := [m] \cup \{0\}$. For $e \in \mathbb{R}^d$ we denote by $e[i]$ the $i$th entry of $e$. Sometimes, we may slightly abuse the notation by indexing the entries by other objects than the integers $i \in [d]$, such as the set of vertices of a graph. We can assume that there is a bijection between these objects and the set $[d]$.

Let $\Sigma$ be a set of class labels; unless stated otherwise, we use $\Sigma = \{\mathsf{blue}, \mathsf{red}\}$. A decision tree in $\mathbb{R}^d$ with set of classes $\Sigma$ consists of an ordered binary tree $T$, that is, each inner node has a well-defined left and right child. Let $\mathsf{feat}: V(T) \to [d]$ and $\mathsf{thr}: V(T) \to \mathbb{R}$ be labelings of each inner node $v \in V(T)$ by a *feature* $\mathsf{feat}(v) \in [d]$ and a *threshold* $\mathsf{thr}(v) \in \mathbb{R}$. Additionally, let $\mathsf{cla}: V(T) \to \Sigma$ be a labeling of the leaves of $T$ by class labels. The tuple $(T, \mathsf{feat}, \mathsf{thr}, \mathsf{cla})$ is a *decision tree* in $\mathbb{R}^d$ with set of classes $\Sigma$. We often omit the labelings $\mathsf{feat}, \mathsf{thr}, \mathsf{cla}$ and just refer to the tree $T$. The *size* of $T$ is the number of its inner nodes, also referred to as *cuts*.

A *training data set* is a tuple $(E, \lambda)$ of a set of *examples* $E \subseteq \mathbb{R}^d$ and their class labeling $\lambda: E \to \Sigma$. Given a training data set, we fix for each feature $i$ a minimum-size set $\mathsf{Thr}(i)$ of thresholds that distinguishes between all values of the examples in the $i$th feature. In other words, for each pair of examples $e$ and $e'$ with $e[i] < e'[i]$, there is at least one value $x \in \mathsf{Thr}(i)$ such that $e[i] < x < e'[i]$. For a feature $i \in [d]$ and a threshold $x \in \mathsf{Thr}(i)$, we use $E_{\leq}[i, x] := \{e \in E : e[i] \leq x\}$ and $E_{>}[i, x] := \{e \in E : e[i] > x\}$ to denote the set of examples of $E$ whose $i$th feature is less or equal, and strictly greater than $x$, respectively.

Now, let $T$ be a decision tree. Each node $v \in V(T)$, including the leaves, defines a subset $E[T, v] \subseteq E$ as follows. For the root $v$ of $T$, we define $E[T, v] := E$. For each non-root node $v$, let $w$ denote the parent of $v$. We then define $E[T, v] := E[T, w] \cap E_{\leq}[\mathsf{feat}(w), \mathsf{thr}(w)]$ if $v$ is the left child of $w$ and $E[T, v] := E[T, w] \cap E_{>}[\mathsf{feat}(w), \mathsf{thr}(w)]$ if $v$ is the right child of $w$. If the tree $T$ is clear from the context, we simplify $E[T, v]$ to $E[v]$. Thus for each example $e \in E$ there is a unique leaf $v$ such that $e \in E[v]$. We also say that $v$ is the *leaf of* $e$. Note that the sets $E[v]$ at the leaves $v$ of $T$ form a partition of $E$. If $v$ is the leaf of $e$, we say that $\mathsf{cla}(v)$ is the class *assigned* to $e$ by $T$. An example $e \in E$ is *correctly classified* by $T$ if the class assigned to it is $\lambda(e)$, and otherwise it is referred to as being *misclassified* or an *error*.

*Identical feature values.* Note that we allow our examples to have identical values in all features, and this occurs in our reductions. However, they could also be adjusted to have no two identical examples, at the cost of increasing $D$ and $\delta_{\max}$:

all our thresholds are integers in the reductions, so changing a value $x$ of some feature to $x'$ with $\lfloor x \rfloor < x' \leq \lceil x \rceil$ will not change the leaf the example ends up at.

*Reasonable trees.* We assume that the input tree $T$ in DTREP, DTRAIS$_=$, DTRAIS$_\geq$ is *reasonable*, that is, (a) no leaf is empty and (b) every leaf has the label of a most frequent example set in this leaf. That is, the set of examples at each cut should be nonempty, and the threshold there should partition that set into two nonempty sets. Note that all trees computed by standard heuristics are reasonable. We make this assumption purely for our hardness results to be more relevant to practical situations. With the replacement operation, since the input trees are reasonable, the number of errors cannot decrease as more cuts are pruned. This is not the case with raising operations (see Theorem 5.4) and thus we study both DTRAIS$_=$ and DTRAIS$_\geq$.

*Relations between parameters.* Note that since we are only interested in the classification properties of subtrees of the input tree $T$, we can omit from all examples all features that do not occur in cuts in $T$. Similarly, we can preprocess the domains in each feature $i \in [d]$: We may look at all the thresholds that occur in feature $i$ in some cut in $T$, say their number is $D'_i$. Then we can discretize the examples to the values in-between such thresholds. Accounting for a minimum and maximum value, we may thus assume that the domain of $i$ contains at most $D'_i + 2$ values. Hence, the maximum domain size $D$ is upper bounded by 2 plus the maximum number of thresholds that occur in a feature in $T$.

Proofs marked with ($\bigstar$) are deferred to the appendix.

## 3. Results for Subtree Replacement

In this section, we present our results for the subtree replacement operation. The problem is solvable in polynomial time with dynamic programming (DP) by a reduction to TREE KNAPSACK, and an improved version of the algorithm requires $\mathcal{O}((n + \ell)s)$ time (Almuallim, 1996). However, if only a small number of cuts are pruned, then the time complexity is quadratic in the size. We propose a novel algorithm whose complexity is only linear in the size if the number of pruned cuts or allowed misclassifications is small:

**Theorem 3.1.** DTREP *can be solved in time* $\mathcal{O}((n + \min\{k^2, t^2\}) \cdot s)$.

*Proof.* First, we compute for each node $v$ the number of misclassified examples $t_v$ in the subtree rooted at $v$ if we were to replace the subtree by a red or a blue leaf, whichever minimizes the number of errors. This requires $\mathcal{O}(ns)$ time in the worst case and less if the tree is not deep. Similarly, we compute the size $s_v$ of the subtree rooted at each $v$.

We use bottom-up dynamic programming, indexing the recurrence by the current node and the number of pruned

nodes or errors, depending on which of $k$ and $t$ is smaller. If $k \le t$, let $\mathrm{opt}(v, k')$ for $k' \in [0, k]$ be the smallest possible number of errors in the subtree rooted at $v$ after pruning exactly $k'$ inner nodes from it. Further, let $u$ and $w$ be the children of $v$. Then, let $\mathrm{opt}(v, s_v - 1) = t_v$ if $s_v - 1 \le k$, and, for $k' < s_v - 1$, let

$$\mathrm{opt}(v, k') = \min_{k^* \in [k']} \mathrm{opt}(u, k^*) + \mathrm{opt}(w, k' - k^*).$$

If $v$ is the root of the tree and $\mathrm{opt}(v, k) \le t$, then there exists a feasible pruned tree for the instance of DTREP. Otherwise, if $t < k$, we use a similar algorithm but instead maximize the number of pruned nodes given that we get $t'$ errors in the subtree. A solution exists if we can prune at least $k$ cuts while having at most $t$ errors, since we can prune fewer cuts without introducing more errors. □

Currently, computing the number of misclassifications in each subtree dominates the time complexity and requires $\mathcal{O}(ns)$ time. We next speed up the classification of examples with heavy–light decompositions (Sleator & Tarjan, 1983), thus obtaining a faster algorithm for DTREP.

**Lemma 3.2 (★).** *After $\mathcal{O}(ds)$-time preprocessing, we can classify any example in time $\mathcal{O}(d \log^2 s)$.*

**Corollary 3.3.** DTREP *can be solved in time* $\mathcal{O}\big(\min\{k^2, t^2\} \cdot s + nd \log^2 s\big)$.

*Proof.* First, classify all $n$ examples by utilizing Lemma 3.2. Second, use the algorithm of Theorem 3.1. □

Interestingly, pruning with subtree replacement becomes hard if we consider *tree ensembles*. A tree ensemble $\mathcal{T}$ is a set of decision trees and $\mathcal{T}$ *classifies* $(E, \lambda)$ if for each example $e \in E$ the majority vote of the trees in $\mathcal{T}$ agrees with the label $\lambda(e)$; ties are broken consistently.

We show this by reducing from the NP-hard $\kappa$-BICLIQUE problem (Johnson, 1987). The constructed ensemble has two trees, one for each partite set. Each tree consists of a long root-to-leaf path that cannot be pruned without violating the error bound. To the unspecified children of this fixed path we attach a cut corresponding to a vertex selection. We then create edge examples $e$ which are correctly classified only if both cuts corresponding to the endpoints of $e$ are preserved. Parameter $\ell$ is chosen such that we can only preserve $2 \cdot \kappa$ cuts. The desired error bound forces us to select exactly $\kappa$ cuts per tree which correspond to a $\kappa$-biclique.

**Theorem 3.4 (★).** DTREP *is NP-hard even for an ensemble of 2 trees where both trees are reasonable and $d = 3$.*

## 4. Algorithms for Subtree Raising

Before presenting our main algorithmic results, note that DTRAIS$_=$ is trivially in XP with respect to $k$ and $\ell$, and

DTRAIS$_\ge$ with respect to $\ell$: we iterate over all $\mathcal{O}(s^k)$ possible combinations of subtrees that are pruned away or $\mathcal{O}(s^\ell)$ combinations of unpruned cuts. FPT for $s$ follows from there being at most $2^s$ possible pruned trees. Thus, both DTRAIS$_=$ and DTRAIS$_\ge$ are also FPT for $k + \ell$. Similarly, since each input decision tree is reasonable, we have $s \le n$ and thus both are FPT for $n$. We start by presenting an XP-algorithm for the number $d$ of features that serves as a starting point for the rest of the algorithms developed in this subsection.

**Theorem 4.1.** DTRAIS$_=$ *and* DTRAIS$_\ge$ *can be solved in* $\mathcal{O}(D^{2d} \cdot s^3)$ *time.*

The algorithm uses bottom-up dynamic programming on the input tree $T$. Intuitively, for each node $v$ of $T$ we compute the minimum number of errors achievable by raising operations that prune at least (or exactly) $k$ nodes in the subtree of $T$ rooted at $v$. In order to do that, we need to be able to determine the set $E'$ of examples that are classified in $v$'s subtree after pruning. Set $E'$ may be different from $E[T, v]$ because in an optimal solution we may have to prune some nodes on the path $P$ in $T$ from the root to $v$. Think of the nodes on $P$ as successively cutting away examples from $E'$, that is, if a node $w$'s successor on $P$ is a left child, $w$ cuts away examples on the left in its feature and if it is a right child, $w$ cuts away examples on the right. To find $E'$ it is thus sufficient, for each feature $i$, to determine the two strongest cuts that remain after raising. That is, among all cuts that cut away examples on the left the strongest cut would be the rightmost one and among all cuts that cut away examples on the right, the strongest cut would be the leftmost one. Therefore we index the table, in addition to $v$ and the remaining budget $k'$, in each feature with the thresholds corresponding to the two strongest remaining cuts.

*Proof of Theorem 4.1.* We only show the result for DTRAIS$_\ge$, the proof for DTRAIS$_=$ is analogous.

**Definition of the DP table:** For each node $v \in V(T)$ let $T_v$ be the subtree of $T$ rooted at $v$. Denote by $E[(\ell_i, r_i)_{i \in [d]}]$ the set of examples in $E$ within the box defined by $(\ell_i, r_i)_{i \in [d]}$ with $\ell_i, r_i \in \mathsf{Thr}(i)$, that is,

$$E[(\ell_i, r_i)_{i \in [d]}] := \bigcap_{i \in [d]} E_>[i, \ell_i] \cap E_\le[i, r_i].$$

We index the DP table $Q$ by the root node $v \in V(T)$ of the subtree, remaining budget $k' \in [0, k]$, and the thresholds $(\ell_i, r_i)_{i \in [d]}$ with $\ell_i, r_i \in \mathsf{Thr}(i)$. To an entry $Q[v, (\ell_i, r_i)_{i \in [d]}, k']$, we put the minimum number of misclassifications achievable on the example set $E[(\ell_i, r_i)_{i \in [d]}]$ with a tree obtained from the subtree $T_v$ by raising operations that prune at least $k'$ inner nodes from $T_v$.

**Location of the solution:** A solution to DTRAIS$_\ge$ can be

read off from $Q$ by letting $v$ be the root of $T$, $k' = k$, and $\ell_i = \min(\mathsf{Thr}(i))$ and $r_i = \max(\mathsf{Thr}(i))$ for each $i$.

**Initialization of $Q$:** The values at a leaf $v$ are the numbers of examples in $E[(\ell_i, r_i)_{i \in [d]}]$ with a different label from $v$, since a leaf cannot be pruned without removing the parent.

**Recurrence of $Q$:** Let $u, w$ be the left and right child of $v$, respectively, and let $s_u, s_w$ be the number of inner nodes in $T_u$ and $T_w$, respectively. We claim that

$$Q[v, (\ell_i, r_i)_{i \in [d]}, k'] =$$
$$\min \begin{cases} Q[u, (\ell_i, r_i)_{i \in [d]}, k' - s_w - 1], \\ Q[w, (\ell_i, r_i)_{i \in [d]}, k' - s_u - 1], \\ \min_{k'' \in [k'] \cup \{0\}} Q[u, (\ell_i, r_i^u)_{i \in [d]}, k''] + \\ \qquad Q[w, (\ell_i^w, r_i)_{i \in [d]}, k' - k''], \end{cases} \quad (1)$$

where $\ell_i^w = \ell_i$ and $r_i^u = r_i$ if $i \neq \mathsf{feat}(v)$, and otherwise $r_{\mathsf{feat}(v)}^u = \min\{r_{\mathsf{feat}(v)}, \mathsf{thr}(v)\}$ and $\ell_{\mathsf{feat}(v)}^w = \max\{\ell_{\mathsf{feat}(v)}, \mathsf{thr}(v)\}$.

**Correctness of the DP:** To see that Equation (1) is correct, we first show that the left-hand side is smaller or equal to the right-hand side. Let $T_v'$ be obtained from $T_v$ by pruning at least $k'$ nodes by raising. There are three cases:

First, $v$, the subtree $T_w$, and possibly some nodes in $T_u$ are pruned to obtain $T_v'$. Note that, then, the number of errors of $T_v'$ for $E[(\ell_i, r_i)_{i \in [d]}]$ is at least $Q[u, (\ell_i, r_i)_{i \in [d]}, k' - s_w - 1]$. Analogously, if $v$ and the subtree $T_u$ are pruned, then the number of errors of $T_v'$ for $E[(\ell_i, r_i)_{i \in [d]}]$ is at least $Q[w, (\ell_i, r_i)_{i \in [d]}, k' - s_u - 1]$.

In the third case, $v$ is not pruned and all raising operations in $T_v$ are contained in $T_u$ and $T_w$. Let $T_u'$ and $T_w'$ be the resulting trees and let $k_u'$ and $k_w'$ be the number of pruned nodes in $T_u$ and $T_w$, respectively. Further, let $t_u$ and $t_w$ be the numbers of misclassifications in $T_v'$ that occur in $T_u'$ and $T_w'$, respectively. Observe that the example set classified by $T_u'$ is $E[(\ell_i, r_i^u)_{i \in [d]}]$, where $r_u$ is defined as in the recurrence. Analogously, the example set classified by $T_w'$ is $E[(\ell_i^w, r_i)_{i \in [d]}]$. Thus, $t_u + t_w \geq Q[u, (\ell_i, r_i^u)_{i \in [d]}, k_u'] + Q[w, (\ell_i^w, r_i)_{i \in [d]}, k_w']$. Hence, the left-hand side of the recurrence equals at most the right-hand side. We defer the other direction to Appendix B.1.

**Running time of the DP:** Observe that there are $s \cdot D^{2d} \cdot s$ table entries, and each entry can be computed in $\mathcal{O}(s)$ time. Proof for DTRAIS$_=$ is analogous but we prune exactly $k$ nodes instead of at least $k$ nodes in the definition of $Q$. $\square$

By only considering thresholds that are actually used in the input tree, we can improve the running time. For this, we define the following parameter: Let $D_T$ be the maximum number of different thresholds on cuts in feature $i$ on path $P$ over all features $i \in [d]$ and all root-to-leaf paths $P$.

**Theorem 4.2 ($\bigstar$).** *DTRAIS$_=$ and DTRAIS$_\geq$ can be solved in $\mathcal{O}((D_T + 2)^{2d_T} \cdot s^3)$ time.*

So far, all of our results applied to both the at least and the exactly variant of subtree raising. However, for DTRAIS$_=$, we can achieve an FPT-algorithm for $k+d$. Later, in Proposition 5.7 we show that such a result for DTRAIS$_\geq$ is unlikely to exist under standard complexity theory assumptions.

**Theorem 4.3.** *DTRAIS$_=$ can be solved in $\mathcal{O}((k+1)^{2d_T} \cdot s^3)$ time.*

*Proof Sketch.* We proceed analogously to Theorems 4.1 and 4.2 for filling a table $Q$ via bottom-up dynamic programming on the input tree $T$. The main difference is that we restrict the possibilities for the values of the thresholds $\ell_i, r_i$. Intuitively, if we focus on the strongest $k + 1$ cuts on the left in a specific feature $i$, at least one them cannot be pruned. Thus, the threshold $\ell_i$ that we index our table with has to be among these $k + 1$ cuts. This restricts the number of table entries to $\mathcal{O}((k + 1)^{2d_T} \cdot s^2)$.

**Definition of the DP table:** Table $Q$ is defined similarly to Theorem 4.1, but we only consider relevant (see below) threshold sequences $(\ell_i, r_i)_{i \in [d]}$ for node $v$ with $\ell_i, r_i \in \mathsf{Thr}(i)$. Intuitively, the thresholds $(\ell_i, r_i)$ must be the thresholds of the strongest cuts in feature $i$ that occur above $v$ in $T$ after removing at most $k - k'$ cuts above $v$. Let $P$ be the path in $T$ from $v$ to the root. For each feature $i \in [d]$ let $(\ell_i^j)_{j \in J_\ell}$ be the list of thresholds of the cuts on the left above $v$ ordered from right to left (largest to smallest). That is, to obtain $(\ell_i^j)_{j \in J_\ell}$, take the set of thresholds of cuts $y$ on $V(P) \setminus \{v\}$ such $\mathsf{feat}(y) = i$ and $y$'s predecessor on $P$ is a right child, and then order it descendingly. Similarly, let $(r_i^j)_{j \in J_r}$ be the list of thresholds of the cuts on the right above $v$ ordered from left to right (smallest to largest).

We now need notation to refer to the strength of a cut $c$, which is intuitively one plus the number of cuts that have to be pruned such that $c$ becomes the strongest cut. For this, let $\mathsf{id}_\ell(i)$ be the index of $\ell_i$ in $(\ell_i^j)_{j \in J_\ell}$, that is, if $\ell_i = \ell_i^j$, then $\mathsf{id}_\ell(i) = j$. Note that the index is well-defined. Analogously, let $\mathsf{id}_r(i)$ be the index of $r_i$ in $(r_i^j)_{j \in J_r}$. Sequence $(\ell_i, r_i)_{i \in [d]}$ is *relevant* (for node $v$) if the remaining budget $k'$ together with the number of cuts that need to be pruned above $v$ such that the thresholds $\ell_i, r_i$ correspond to the strongest cuts do not exceed $k$. Formally, it must hold that $k \geq k' + \sum_{i \in [d]}(\mathsf{id}_\ell(i) + \mathsf{id}_r(i) - 2)$. Observe that the sum indeed measures the number of nodes we have to remove from $P$ so that the strongest remaining cuts have the thresholds specified in $(\ell_i, r_i)_{i \in [d]}$.

**Recurrence and correctness of the DP:** The same as in Equation (1), except that we will not prune $v$ if it would lead to too many cuts being removed to obtain the specified thresholds. Moreover, we will not prune the

cut if $k < k' + \sum_{i \in [d]}(\mathsf{id}_\ell(i) + \mathsf{id}_r(i) - 2)$. Note that, in this way, if the sequence $(\ell_i, r_i)_{i \in [d]}$ is relevant for $v$ then also in the table entries on the right-hand side it is the case that the sequences of thresholds are relevant for the corresponding nodes. We omit a correctness proof of the recurrence because it is analogous to the proof of Thm. 4.1.

**Running time:** There are $(k+1)^{2d_T} s^2$ relevant threshold sequences, and each entry is computed in $\mathcal{O}(s)$ time. $\qquad\square$

Next, we prove that DTRAIS$_\geq$ is in XP for $k + t$. We start with the special case of $k = t = 0$:

**Lemma 4.4 (★).** DTRAIS$_\geq$ *can be solved in* $\mathcal{O}(ns)$ *time if* $k = t = 0$.

**Theorem 4.5 (★).** DTRAIS$_\geq$ *is solvable in* $\mathcal{O}(n^{t+1}s^{k+1})$ *time.*

**Theorem 4.6 (★).** DTRAIS$_\geq$ *can be solved in* $\mathcal{O}((6\delta_{\max}D\ell(t+1))^\ell \cdot \ell^2 ns)$ *time.*

Note that since for DTRAIS$_=$ we cannot assume that the pruned tree is minimal, we cannot use the algorithm of Theorem 4.6.

## 5. Hardness Results for Subtree Raising

In this subsection we show by complementing hardness results that our algorithmic results from Section 4 cannot be improved substantially without violating standard complexity assumptions. Specifically, we use the Exponential Time Hypothesis (ETH), which states that states that 3SAT on $n$-variable formulas cannot be solved in $2^{o(n)}$ time (Impagliazzo & Paturi, 2001; Impagliazzo et al., 2001). In this section, by $I$ we denote the instance of DTRAIS$_=$ or DTRAIS$_\geq$ we construct in the reductions.

First, we show that the trivial brute-force algorithm for DTRAIS$_=$ cannot be improved significantly.

**Theorem 5.1.** *Even if* $\delta_{\max} = 2$, $D = 2$, *and* $t = 0$ *both* DTRAIS$_=$ *and* DTRAIS$_\geq$ *are* W[1]-*hard for* $k$ *and, unless the ETH is false, they cannot be solved in* $f(k)\cdot|I|^{o(k)}$ *time.*[5]

*Proof.* We prove the statement by a reduction from the W[1]-hard (Downey & Fellows, 1995) problem of $\kappa$-INDEPENDENT SET which cannot be solved in $f(\kappa) \cdot n^{o(\kappa)}$ time unless the ETH fails (Cygan et al., 2015). The input is a graph $G$ and an integer $\kappa$, and the task is to find a set $S \subseteq V(G)$ of size at least $\kappa$ such that no edge of $G$ has both endpoints in $S$. With slight abuse of notation, we let the vertices of $G$ be the (binary) features of the examples. We first show the statement for non-reasonable trees. We

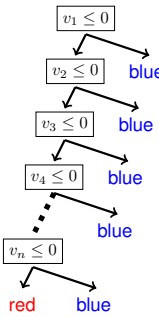

*Figure 4.* The initial decision tree used for Theorem 5.1.

construct a decision tree $T$ whose inner nodes relate to the vertices of $G$ in the sense that pruning a subset of inner nodes leads to no misclassifications if and only if the corresponding set of vertices of $G$ is an independent set. Further, we let $k = \kappa$, and $t = 0$ in our reduction.

We create one blue example $e$ for each edge $(v, w)$ of $G$ such that $e[v] = e[w] = 1$ and otherwise $e[u] = 0$ for $u \in V(G)$. Additionally, we create one red example $e$ with $e[v] = 0$ for all $v \in V(G)$. The inner nodes of our initial decision tree $T$ form a path such that there is a single cut with respect to each feature $v \in V(G)$ in an arbitrary but fixed order. If $e[v] = 1$, then the example is directed to a blue leaf, and otherwise passed forward on the path. At the end of the path, there is a red leaf. This is illustrated in Figure 4.

Initially, all examples are classified correctly. Any subtree including the unique red leaf cannot be pruned, because then the unique red example would be misclassified. Consequently, each raising operation removes an inner node and the blue leaf attached to it. On the other hand, the example corresponding to an edge $(v, w)$ gets misclassified if the cuts related to vertices $v$ and $w$ are both pruned. Hence, the pruned features correspond an independent set in $G$, and, if exactly/at least $k$ cuts can be pruned, then there exists an independent set of size exactly/at least $k$. Since $k = \kappa$ we obtain the $f(k)\cdot|I|^{o(k)}$ time lower bound if the ETH is true. We defer adaption for reasonable trees to the Appendix. $\qquad\square$

Now, we show, by adapting the proof of Theorem 5.1, that also the simple $\mathcal{O}(s^\ell)$ time brute-force algorithms for DTRAIS$_=$ and DTRAIS$_\geq$ cannot be improved significantly.

**Theorem 5.2 (★).** *Even if* $\delta_{\max} = 2$ *and* $D = 2$, *both* DTRAIS$_=$ *and* DTRAIS$_\geq$ *are* W[1]-*hard for* $\ell$ *and, unless the ETH is false, they cannot be solved in* $\mathcal{O}(f(\ell) \cdot |I|^{o(\ell)})$ *time, even if* $\delta_{\max} = 2$, *and* $D = 2$.

**Theorem 5.3 (★).** *Even if* $t = 0$ *and* $D = 2$, *both* DTRAIS$_=$ *and* DTRAIS$_\geq$ *are* W[2]-*hard for* $\ell$ *and, unless the ETH is false, they cannot be solved in* $\mathcal{O}(f(\ell) \cdot |I|^{o(\ell)})$ *time, even if* $t = 0$ *and* $D = 2$.

Next, we show that DTRAIS$_\geq$ is substantially harder than

---

[5]The Exponential Time Hypothesis (ETH) states that 3-SAT on $n$-variable formulas cannot be solved in $2^{o(n)}$ time, see Impagliazzo & Paturi (2001); Impagliazzo et al. (2001) for details.

DTRAIS$_=$ with respect to $k$: A similar $\mathcal{O}(n^k)$ time brute-force algorithm for DTRAIS$_\geq$ implies P=NP. This result is based on the observation that the number of errors does not increase monotone if more raising operations are performed.

**Theorem 5.4 (★).** *For every positive integer $k$ there is a training data set $(E, \lambda)$ and an initial decision tree $T$ with zero errors such that only performing $k$ raising operations leads to a tree without errors and performing $j$ raising operation for any $1 \leq j < k$ leads to at least one error.*

**Theorem 5.5 (★).** DTRAIS$_\geq$ *is NP-hard even if $k = 0$, $\delta_{\max} = 2$, and $D = 2$.*

We now show that our XP-algorithm for $d$ (Theorem 4.1) cannot be improved to an FPT-algorithm and that the exponential dependence on $d$ cannot be reduced substantially.

Intuitively, we create two features $d_i^<$ and $d_i^>$ per color class $i$ such that in the pruned tree we need to preserve exactly one cut in each feature to fulfill the desired error bound. We achieve this property by adding a huge number of blue examples and red examples which can only be distinguished in $d^*$. The two cuts in features $d_i^<$ and $d_i^>$ force us to select exactly one vertex $v_i$ of this color class. For each edge we create an *edge example*. If vertex $v_i$ is selected, then all edge examples corresponding to edges having an endpoint in color class $i$ which is *not* $v_i$ are then misclassified. Thus, we can only correctly classify an edge example if we select both endpoints of that edge. Furthermore, we ensure that only edge examples may be misclassified without violating the error bound. Hence, by setting $t := m - \binom{\kappa}{2}$, we ensure that we need to select a multicolored clique.

**Theorem 5.6 (★).** *Even if $\delta_{\max} = 6$, both DTRAIS$_=$ and DTRAIS$_\geq$ are W[1]-hard for $d + \ell$ and, unless the ETH is false, they cannot be solved in $f(d + \ell) \cdot |I|^{o(d+\ell)}$ time.*

Recall that in Theorem 4.3 we showed that DTRAIS$_=$ is FPT with respect to $k + d$. By adapting the proof of Theorem 5.6 slightly, we show that this is unlikely for DTRAIS$_\geq$.

**Proposition 5.7 (★).** DTRAIS$_\geq$ *is W[1]-hard for $d$ even if $k = 0$ and $\delta_{\max} = 6$.*

Finally, we show that the combination of $d$ and $t$ is unlikely to yield an FPT-algorithm.

**Theorem 5.8 (★).** *Even if $t = 0$ and $\delta_{\max} = 6$, both DTRAIS$_=$ and DTRAIS$_\geq$ are W[1]-hard for $d$ and, unless the ETH is false, they cannot be solved in $f(d) \cdot |I|^{o(d)}$ time.*

## 6. Experiments

The main goal of our small-scale empirical study was to study whether common heuristics for decision-tree pruning find optimal tradeoffs between the number of pruned nodes and errors on the resulting trees. In other words, we assess if the heuristics achieve near-maximum number of pruned

*Table 1.* Datasets with improvable heuristic results: $s$ is the initial unpruned tree size, $k_{\text{rais}}$ the number of pruned nodes by the raising heuristic, $k_{\text{repl}}$ the number of pruned nodes by the replacement heuristic, $k^*$ the maximum number of nodes that can be pruned by raising operations while maintaining at most $t_{\text{rais}}$ errors. Column $t_{\text{rais}}$ contains the number of errors obtained by the raising heuristic, $t_{\text{repl}}$ the number of errors obtained by the replacement heuristic, and $t^*$ the minimum number of errors obtainable by pruning at least $k_{\text{rais}}$ nodes with raising operations.

| Dataset | $s$ | $k_{\text{rais}}$ | $k_{\text{repl}}$ | $k^*$ | $t_{\text{rais}}$ | $t_{\text{repl}}$ | $t^*$ |
|---|---|---|---|---|---|---|---|
| soybean | 28 | 15 | 15 | 17 | 8 | 8 | 7 |
| cleveland-nominal | 46 | 38 | 38 | 39 | 23 | 23 | 22 |
| haberman | 92 | 74 | 71 | 75 | 39 | 38 | 38 |
| postoperative-patient | 23 | 3 | 2 | 3 | 1 | 1 | 1 |
| heart-statlog | 54 | 31 | 27 | 31 | 17 | 15 | 17 |

cuts for the chosen number of errors, and near-minimum number of errors for the chosen number of pruned cuts.

We used 40 datasets from the Penn Machine Learning Benchmarks library (Romano et al., 2022), including 32 previously used for minimum-size tree computation (Bessiere et al., 2009; Narodytska et al., 2018; Staus et al., 2025)[6] and additional larger datasets. The datasets range from 72 to 5404 examples (mean 674.88, median 302). We computed unpruned and pruned trees using WEKA 3.8.5's (Frank et al., 2010) C4.5 implementation (Quinlan, 1993): The unpruned trees apply neither replacement nor raising, whereas pruned trees used only the replacement or raising heuristics; more details in the appendix (in particular, Table 2).

Analysis of the unpruned trees showed that several parameters are suitably small with medians $s\colon 26, d\colon 9, D\colon 6$. We thus used a dynamic-programming algorithm based on Theorem 4.2; the source code and data are publicly available (Harviainen et al., 2025b). The implementation is a proof of concept and has lots of room for optimizations. We computed the Pareto front that contains for each number $k$ of pruned nodes via subtree raising, the minimum-possible classification error of the resulting pruned tree. With memory limit of 64GB and time limit of 24h this was achieved for 26 instances.

Key questions we can now answer (see also Table 1):

- Do heuristics achieve close to minimum-possible errors for their chosen number of pruned nodes? Mostly yes: In only four of the 26 solved instances, the heuristics did not achieve the minimum-possible errors.
- Do heuristics prune close to maximum number of nodes for their chosen number of errors? Mostly yes: In only four instances the heuristics are suboptimal.

---

[6]Staus et al. (2025) used 35 data sets. We had to exclude 3 of them (auto, cloud, spect) since for these dataset the heuristics only outputted a tree with a single cut.

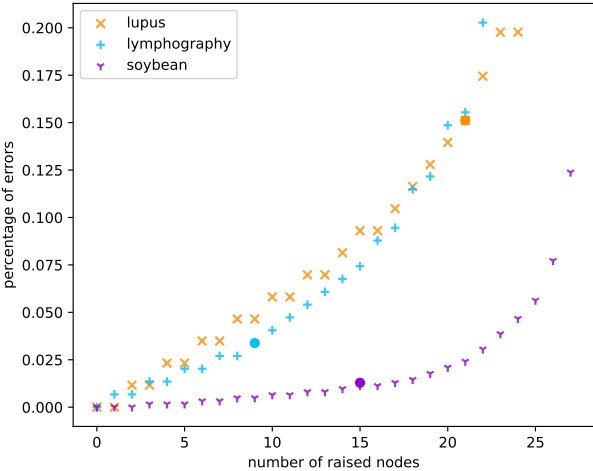

*Figure 5.* The trade-off between number of raising operations and classification error for three instances. The thin points are the optimal values computed by our algorithm and the thick circles are the values achieved by the raising heuristic of WEKA.

For a visualization of the Pareto front for three instances, see Figure 5. Soybean is one of the few instances where the number of errors of the heuristic is larger than the optimum for that number of raising operations. One can see that in these instances more nodes can be pruned than the heuristics do without significantly worsening the accuracy.

The study indicates that commonly used heuristics perform well in most cases, with only a few instances showing room for improvement in either error minimization or node pruning maximization. However, inspection of the Pareto front leads us to hypothesize that often more nodes can be pruned than the heuristics do without worsening the accuracy significantly.

## 7. Outlook

We provided a comprehensive analysis of the parameterized complexity of optimal pruning with subtree replacement and subtree raising, presenting algorithmic results and complexity-theoretic lower bounds for each operation. Further, we performed a small-scale experiment, showing the surprising result that pruning heuristics are almost optimal despite the hardness of the problem, that is, in almost every of our data sets no smaller classification error can be achieved by pruning the same number of nodes. Our algorithms were crucial for discovering this, since without them we could not compare the heuristics against the optimum.

In our paper we solely focused on the biclass setting for clarity of the theory. All our hardness results directly apply to the multiclass setting. All our algorithms except the one

in Theorem 4.6 also directly apply to the multiclass setting without changes. To adapt the algorithm from Theorem 4.6 we additionally only need to set the leaf to the class of the dirty example, when we introduce a new leaf.

While we managed to determine the parameterized complexity of most parameter combinations, some combinations of at least three parameters remain open, such as whether $\mathrm{DTR_{AIS_=}}$ or $\mathrm{DTR_{AIS_\geq}}$ is FPT with respect to $d + t + \ell$. Regarding optimally pruning ensembles, we showed it to be NP-hard already for two trees. However, the reduction does not show parameterized hardness and it would thus be interesting to see non-trivial fixed-parameter tractability results for important parameters here. This could be a potentially valuable research direction, since ensembles are typically more accurate than a single decision tree of the same size.

More generally, a natural follow-up question is whether we can perform other local operations on the input decision trees where we can beat heuristics more clearly. A promising candidate in this direction could be an operation that can arbitrarily reconstruct parts of the decision tree (Schidler & Szeider, 2024), thus enabling local changes without removing entire subtrees. Another direction is to investigate related methods such as local search (Carreira-Perpiñán & Tavallali, 2018; Saremi & Yaghmaee, 2018). What is the complexity of computing the associated operations optimally and exhaustively?

## Acknowledgements

Juha Harviainen was supported by the Research Council of Finland, Grant 351156. Frank Sommer was supported by the Alexander von Humboldt Foundation. Stefan Szeider was supported by the Austrian Science Fund (FWF) within the projects 10.55776/COE12 and 10.55776/P36420.

## Impact Statement

This paper presents work whose goal is to advance the field of Machine Learning. There are many potential societal consequences of our work, none which we feel must be specifically highlighted here.

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

# Appendix

# A. Additional Material for Section 3

### A.1. Proof of Lemma 3.2

*Proof.* We say that an edge $(v, u)$ from $v$ to $u$ in a decision tree $T$ is *heavy* if the number of nodes in the subtree rooted at $v$ is less than twice the number of nodes that the subtree rooted at $u$ has. Otherwise, edge $(v, u)$ is *light*. Now, any root-to-leaf path has at most $\mathcal{O}(\log s)$ light edges, and the graph edge-induced by the heavy edges is a disjoint union of paths (Sleator & Tarjan, 1983). Constructing this heavy–light decomposition takes $\mathcal{O}(s)$ time.

Now, we compute for each cut $v$ and feature the tightest lower and upper bounds with respect to that feature on the path from the root to $v$. In other words, we precompute a table for characterizing the interval of values on a feature which an example can potentially have if it ends up at that cut. This takes $\mathcal{O}(ds)$ time.

Suppose now that we want to classify example $e$. Since the edge-induced subgraph of the heavy edges is a disjoint union of paths, every cut belongs to a unique heavy path, possibly of length 0. Now, let $r$ be the root ot $T$ and let $P_r$ be the unique heavy path containing $r$. Next, we compute how far $e$ goes on the heavy path $P_r$ by binary search: $e$ can only end up at a cut of $P_r$ if the value of each feature falls in the interval of possible values we precomputed for all cuts. Testing this for a single cut takes $\mathcal{O}(d)$ time and the binary search thus takes $\mathcal{O}(d \log s)$ time. Then, the example goes trough a light edge to another heavy path $P_s s$, and we continue with a binary search on that heavy path $P_s$, repeating the process until we end up at a leaf. Since $T$ contains at most $\mathcal{O}(\log s)$ light edges, we conclude that on every root-to-leaf-path of $T$ there are at most $\mathcal{O}(\log s)$ heavy paths on any root-to-leaf path of $T$. Thus, this process takes $\mathcal{O}(d \log^2 s)$ time in total. $\square$

### A.2. Proof of Theorem 3.4

*Proof.* We reduce from the NP-hard $\kappa$-BICLIQUE problem (Johnson, 1987). The input is a bipartite graph $G$ with partite sets $P = \{p_1, \ldots, p_N\}$ and $Q = \{q_1, \ldots, q_N\}$, $M$ edges, and an integer $\kappa$ such that $G$ has no isolated vertices. The task is to decide whether $G$ contains a complete bipartite subgraph with $\kappa$ vertices on each side.

**Outline:** The idea is to create an ensemble consisting of two trees, one tree for each partite set. Each of these trees consists of a long root-to-leaf path which cannot be pruned without violating the error bound, denoted as a *required path*. To the unspecified children of the required path we attach a further cut which corresponds to a vertex selection. Furthermore, we create edge examples $e$ which can only be correctly classified if both cuts corresponding to the

endpoints of $e$ are preserved. Parameter $\ell$ is chosen such that we can only preserve $2 \cdot \kappa$ cuts, Furthermore, the desired error bound forces us to select exactly $\kappa$ cuts per tree which correspond to a $\kappa$-biclique.

**Construction:** *Description of the data set:* We set blue to the *dominant label*, that is, if some example $e$ is classified as blue by one tree in the ensemble and as red by the other tree in the ensemble, then $e$ is classified as blue. A visualization is shown in Figure 6. Let $S \in \{P, Q\}$ be any partite set.

- For each edge $\{p_i, q_j\} \in E(G)$ we add an *edge example* $e(p_i, q_j)$. To all these examples we assign label red.

- For each $i \in [N]$ and each partite set $S$, we add a set $B_S^i$ of *separation examples*. Each of these sets consists of $M$ examples having the same value in each feature. To all these examples we assign label blue.

- For each partite set $S$, we add a set $B_S$ of blue *forcing examples*. Both of these sets contain exactly $M$ examples and all examples in one of these sets have the same value in each feature.

- For each partite set $S$, we add a set $R_S$ of red *enforcing examples*. Both of these sets contain exactly $4 \cdot N \cdot M$ examples and all examples in one of these sets have the same value in each feature.

We add three features $d_P, d_Q$, and $d_E$. It remains to describe the coordinates of all examples in these features.

- For each edge example $e = e(p_i, q_j)$ we set $e[d_P] = i$, $e[d_Q] = j$, and $e[d_E] = 1$.

- For each separation example $e \in B_P^i$ we set $e[d_P] = i$, and $e[d_Q] = 0 = e[d_E]$. Similarly, for each separation example $e \in B_Q^i$ we set $e[d_Q] = i$, and $e[d_P] = 0 = e[d_E]$.

- For each forcing example $e \in B_P$ we set $e[d_P] = N + 1$, $e[d_Q] = 0$, and $e[d_E] = 1$. Similarly, for each forcing example $e \in B_Q$ we set $e[d_P] = 0$, $e[d_Q] = N + 1$, and $e[d_E] = 1$.

- For each enforcing example $e \in R_P$ we set $e[d_P] = N + 1$ and $e[d_Q] = 0 = e[d_E]$. Similarly, for each enforcing example $e \in R_Q$ we set $e[d_Q] = N + 1$ and $e[d_P] = 0 = e[d_E]$.

*Description of the input ensemble $\mathcal{T}$:* The ensemble $\mathcal{T}$ consists of two trees $T_P$ and $T_Q$. We only describe $T_P$. To obtain $T_Q$, each cut in $d_P$ is replaced by the identical cut in $d_Q$, that is, $d_P \leq x$ is replaced by $d_Q \leq x$.

a)

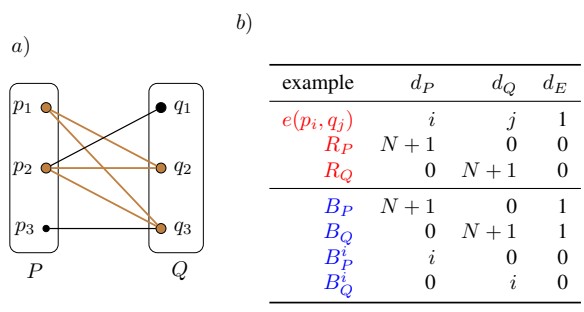

b)

| example | $d_P$ | $d_Q$ | $d_E$ |
|---|---|---|---|
| $e(p_i, q_j)$ | $i$ | $j$ | 1 |
| $R_P$ | $N+1$ | 0 | 0 |
| $R_Q$ | 0 | $N+1$ | 0 |
| $B_P$ | $N+1$ | 0 | 1 |
| $B_Q$ | 0 | $N+1$ | 1 |
| $B_P^i$ | $i$ | 0 | 0 |
| $B_Q^i$ | 0 | $i$ | 0 |

c)

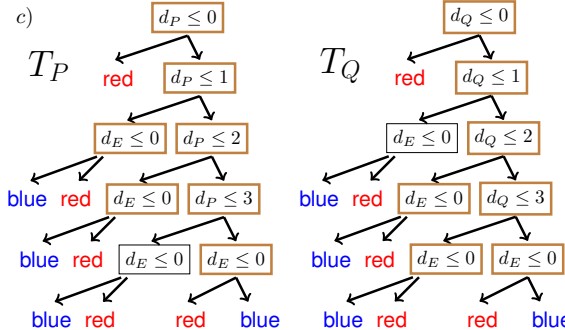

**Figure 6.** A visualization of the reduction from the proof of Theorem 3.4. a) shows a $\kappa$-BICLIQUE instance; a $\kappa$-biclique is depicted in brown. b) shows the corresponding classification instance. c) shows both trees $T_P$ and $T_Q$ of the input ensemble $\mathcal{T}$. The brown cuts correspond to cuts which are preserved in the solution ensemble $\mathcal{T}'$.

One root-to-leaf path of $T_P$ consists of the cuts $d_P \leq 0, d_P \leq 1, \ldots, d_P \leq N, d_E \leq 0$. We call this the *required path* of $T_P$. The left child of the last cut is a red leaf and its right child is a blue leaf. Furthermore, the left child of the first cut is a red leaf. The left child of each remaining cut $d_P \leq i$ for each $i \in [N]$ is the cut $d_E \leq 0$ and its left child is a blue leaf and its right child is a red leaf. This cut is referred to as *the $p_i$-cut*. In $T_Q$ these cuts are denoted as *the $q_i$-cuts*. Since blue is the dominant label, in $\mathcal{T}$ each example is correctly classified. Finally, we set $\ell := 2 \cdot (N+2) + 2 \cdot \kappa$ and $t := M - \kappa^2$.

Clearly, this corresponding instance of DTREP can be constructed in polynomial time. Furthermore, observe that both trees $T_P$ and $T_Q$ of the ensemble are reasonable since $G$ contains no isolated vertices.

**Correctness:** We show that $G$ has a $\kappa$-biclique if and only if $\mathcal{T}$ can be pruned by replacement operations such that the resulting ensemble $\mathcal{T}'$ has exactly $\ell$ inner nodes and makes at most $t$ errors.

($\Rightarrow$) Let $P' \subseteq P$ and $Q' \subseteq Q$ be a $\kappa$-biclique (for example see part a) of Figure 6). To obtain $\mathcal{T}'$, we preserve the required paths of $T_P$ and $T_Q$. Furthermore, for each $p_i \in P'$ we also preserve the $p_i$-cut, that is, the cut $d_E \leq 0$ which is the left child of the cut $d_P \leq i$. Analogously, for each $q_i \in Q'$ we also preserve the $q_i$-cut. In other words, we prune exactly $N - \kappa$ many $p_i$-cuts (where $p_i \notin P'$) and exactly $N - \kappa$ many $q_j$ cuts (where $q_j \notin Q'$). For an example, see part c) of Figure 6. Furthermore, observe that in both the most-frequent tree replacement and the most-frequent ensemble replacement, the label of each new leaf is blue. By $T_P'$ and $T_Q'$ we denote the pruned trees.

Observe that $\mathcal{T}'$ contains exactly $2 \cdot (N+2) + 2 \cdot \kappa = \ell$ cuts. It remains to verify that $\mathcal{T}'$ makes at most $t = M - \kappa^2$ errors.

Since the classification path of each forcing and enforcing example in $T_P'$ and $T_Q'$ stays the same as in $T_P$ and $T_Q$, respectively, all these examples are still correctly classified.

Furthermore, each separation example $e \in B_P^i$ is classified as blue in $T_P'$: either its classification path is not changed or the last cut $d_E < 0$ which is a $p_i$-cut for some $p_i \notin P'$ is pruned and it is replaced by a blue leaf. Since blue is the dominant label, $e$ is correctly classified by $\mathcal{T}'$. An analog argument applies for each separation example $e \in B_Q^i$.

To verify the desired error bound it remains to show that at least $\kappa^2$ edge examples are correctly classified by $\mathcal{T}'$. More precisely, we show that each edge example corresponding to an edge $\{p_i, q_j\}$ in the $\kappa$-biclique is correctly classified by $\mathcal{T}'$. Observe that for each $p_i \in P'$ the classification path of all edge examples $e(p_i, q_z)$, where $q_z$ is a neighbor of $a_i$, in $\mathcal{T}'$ is identical to the one in $\mathcal{T}$. Thus, $e(p_i, q_z)$ is classified as red in $T_P'$. An analog argument applies for $q_i \in Q'$. Thus, each edge example corresponding to an edge of the $\kappa$-biclique is correctly classified by $\mathcal{T}'$. Since any $\kappa$-biclique contains exactly $\kappa^2$ edges the statement follows.

($\Leftarrow$) Let $\mathcal{T}'$ with trees $T_P'$ and $T_Q'$ be a solution for the raising problem, that is, $\mathcal{T}'$ has $\ell = 2 \cdot (N+2) + 2 \cdot \kappa$ inner nodes and makes at most $M = \kappa^2$ errors.

*Outline:* First, we show that in both trees the required paths need to be preserved to fulfill the error bound. Second, we show that any edge example can only be correctly classified by $\mathcal{T}'$ if its classification path in $T_P'$ and $T_Q'$ is identical to the one in $T_P$ and $T_Q$, respectively. Finally, we verify that all correctly classified edge examples correspond to a $\kappa$-biclique.

*Step 1:* If the last cut $d_E \leq 0$ of the required path of $T_P$ is pruned, then the classification paths of all forcing examples in $B_P$ and all enforcing examples in $R_P$ (independent of all other pruning operations) is identical in $\mathcal{T}'$. Since both sets have size at least $M$, $\mathcal{T}'$ would have at least $M$ errors, a contradiction. Since pruning any ancestor of this cut implies also pruning this cut, the entire required path of $T_P$ is not pruned in $T_P'$. Analogously, we can show that the required path of $T_Q$ cannot be pruned.

*Step 2:* Step 1 implies that only $p_i$-cuts of $T_P$ and $q_i$-cuts of $T_Q$ can be pruned. Furthermore, observe that independent of whether the most-frequent tree replacement or the most-frequent ensemble replacement is used, each new leaf which replaces one of these cuts has label blue. Thus, any edge example $e = e(p_i, q_j)$ where $p_i \notin P'$ ends up in a blue leaf in $T'_P$. Since blue is the dominant label, $e$ is misclassified as blue by $\mathcal{T}'$, independent of the classification result of $e$ in $T'_Q$. An analogous argument holds for $q_j \notin Q'$ and $T'_Q$. Thus, $e = e(p_i, q_j)$ is correctly classified if and only if $p_i \in P'$ and $q_j \in Q'$.

*Step 3:* By the definition of $t$, $\mathcal{T}'$ has to classify at least $\kappa^2$ edge examples correctly. Assume that $x$ many $p_i$-cuts of $T_P$ are not pruned and that $z$ many $q_i$ cuts of $T_Q$ are not pruned. By $P'$ and $Q'$ we denote the vertices of $P$ and $Q$ which correspond to the not pruned $p_i$-cuts and $q_i$-cuts, respectively. Note that $x + z = 2 \cdot \kappa$. Since there is at most one edge example for each pair of vertices from $P$ and $Q$, $\mathcal{T}'$ can classify at most $x \cdot z$ edge examples correctly. Hence, we obtain that $x = \kappa = z$. Furthermore, for each $p_i \in P'$ and each $q_j \in Q'$ graph $G$ has to contain the edge $\{p_i, q_j\}$ to fulfill the error bound $t$. Thus, $(P', Q')$ is a $\kappa$-biclique in $G$. $\square$

## B. Additional Material for Section 4

### B.1. Missing material from Theorem 4.1

Now we show that the right-hand side is smaller than or equal to the left-hand side. Consider a tree $T'_u$ obtained from $T_u$ corresponding to $Q[u, (\ell_i, r_i)_{i \in [d]}, k' - s_w - 1]$. Note that pruning $v$ and the subtree $T_w$ from $T_v$, and then performing the raising operations in $T'_u$ yields a tree $T'_v$ that misclassifies exactly $Q[u, (\ell_i, r_i)_{i \in [d]}, k' - s_w - 1]$ examples of $E[(\ell_i, r_i)_{i \in [d]}]$. Furthermore, at least $k'$ nodes have been pruned from $T_v$ to obtain $T'_v$. Hence the right-hand side is smaller or equal to $Q[u, (\ell_i, r_i)_{i \in [d]}, k' - s_w - 1]$. By an analogous argument for $T_u$ the right-hand side is also smaller or equal to $Q[w, (\ell_i, r_i)_{i \in [d]}, k' - s_u - 1]$. Let $t'' = Q[u, (\ell_i, r_i^u)_{i \in [d]}, k''] + Q[w, (\ell_i^w, r_i)_{i \in [d]}, k' - k'']$ wherein $k''$ minimizes the sum. Consider the trees $T'_u$ and $T'_w$ corresponding to $t''$, obtained by raising operations from $T_u$ and $T_w$. Perform the same operations as in $T'_u$ and $T'_w$ in $T_v$ to obtain $T'_v$. Note that $v$ is not pruned. Therefore, the examples of $E[(\ell_i, r_i)_{i \in [d]}]$ classified in the $T'_u$-subtree of $T'_v$ are exactly $E[(\ell_i, r_i^u)_{i \in [d]}]$ and analogously for $T'_w$. Thus, the number of misclassifications in $T'_v$ on $E[(\ell_i, r_i)_{i \in [d]}]$ is exactly $t''$. Hence, the right-hand side of the recurrence is smaller or equal to the left-hand side.

### B.2. Proof of Theorem 4.2

*Proof Sketch.* We use the almost the same definition of the table $Q$ as in the proof of Theorem 4.1: Instead of defining

the table $Q[v, (\ell_i, r_i)_{i \in [d]}, k']$ for all sequences $(\ell_i, r_i)_{i \in [d]}$ of thresholds with $\ell_i, r_i \in \mathsf{Thr}(i)$, instead we restrict these sequences as follows: First, we vary only the thresholds for the features that occur on the path $P$ from the root to $v$ and for all remaining features $i$ we set the thresholds to the fixed maximum and minimum value, respectively. Second, in each feature $i \in [d]$ in which we vary thresholds, we consider not all threshold values $\ell_i, r_i \in \mathsf{Thr}(i)$, but only those at most $D_T$ values that occur on cuts in feature $i$ on $P$ and the minimum and maximum value in feature $i$. To see that the recurrence works in the same way, note that, if the left-hand side is so restricted, then all table entries that we refer to on the right-hand side are also restricted in this way for their corresponding tree nodes. Thus we refer only to table entries that have previously been computed. $\square$

### B.3. Proof of Lemma 4.4

*Proof.* Observe that if an example is misclassified, it will remain misclassified unless the leaf to which it ends up gets pruned away. Consequently, to achieve zero errors, we have to repeatedly prune any leaf containing a misclassified example while such leaves remain. If the whole tree gets pruned, there is thus no solution. During the execution of the algorithm, each example can pass each edge of the decision tree twice, resulting in $\mathcal{O}(ns)$ total work. $\square$

### B.4. Proof of Theorem 4.5

*Proof.* Assume a solution exists. If we prune $k$ cuts from the set of pruned cuts and remove the misclassified examples of the solution, then the algorithm from Lemma 4.4 finds a solution to this reduced instance with $k = t = 0$. Conversely, if no solution exists, then no reduced instance does has a solution either. Iterating over all subsets of cuts of size $k$ and subsets of examples of size $t$ results in the desired time complexity. $\square$

### B.5. Proof of Theorem 4.6

*Proof.* We prove the theorem by exploiting the witness tree algorithm of Komusiewicz et al. (2023b, Section 6.3). They present a method for enumerating decision trees with at most $\ell$ cuts with at most $t$ errors in $\mathcal{O}((6\delta_{\max} D\ell(t + 1))^\ell \cdot \ell n)$ time. Roughly speaking, they utilize the concept of a witness tree where a decision tree is associated with a function that associates one example ending up at each leaf as the *witness* of that leaf. The algorithm starts with a decision tree with only a single leaf node. If there are more than $t$ misclassifications, they arbitrarily pick a subset of $t + 1$ errors. Since at least one of them has to be correctly classified, with branching, they obtain an element $e$ which needs to be correctly classified. Let $v$ be the current leaf of $e$. Now, a new cut is added to the tree that separates $e$ from the witness of $v$. By exploiting the fact that $e$ and the

witness of $v$ differ in at most $\delta_{\max}$ features and each feature has at most $D$ thresholds, there are only $\delta \cdot D$ possibilities for the cut. This process is then repeated at most $\ell$ times. For more details, we refer the reader to the original work (Komusiewicz et al., 2023b).

For each tree $P$ enumerated by the algorithm, we need to test whether $P$ can be obtained from the input decision tree $T$ by pruning operations. If the roots of $T$ and $P$ are identical, then we keep the root of $T$ and recursively continue to find $P_{\text{left}}$, the left subtree of $P$ in $T_{\text{left}}$, the left subtree of $T$, and perform this analogously with the right subtree. If the roots are not identical, then we return the logical OR of finding $P$ in the left subtree or the right subtree of $T$. This adds an additional factor of $\mathcal{O}(\ell s)$ to the running time, making it $\mathcal{O}((6\delta_{\max}D\ell(t+1))^\ell \cdot \ell^2 ns)$ in total.

**Correctness:** If a solution exists, then there is a decision tree $P$ with (at most) $\ell$ cuts that makes at most $t$ errors. One of the decision trees enumerated by the witness tree algorithm of Komusiewicz et al. (2023b) is thus $P$. If no solution exists, the enumeration algorithm may still list some decision trees on (at most) $\ell$ cuts that make at most $t$ errors, but none of them can be obtained from the input decision tree $T$ by pruning operations.

Therefore, we need to show that our algorithm for testing whether $P$ can be obtained from $T$ works correctly. We prove this by induction. The base case of $s = 1$ is trivial. Suppose now the correctness for all $s < s'$; we next show the correctness for $s = s'$.

If $P$ can be obtained from $T$ by pruning operations, then $T$ has a cut that is equal to the root of $P$. If $T$ does not have a cut equal to the root of $P$, then the algorithm correctly outputs that $P$ cannot be obtained from $T$. Assume now that $T$ has such a cut. If it is the root of $T$, then no other cut of $T$ can equal the root of $P$ because of reasonability. Therefore, we cannot prune the root, and the problem reduces to testing whether left (right) subtree of $P$ can be obtained from the left (right) subtree of $T$ by pruning operations. By the induction assumption, the algorithm performs this correctly.

Now assume instead that $T$ has such a cut but it is not the root of $T$. Then, we need to prune the root of $T$ and consequently also one of its subtrees. If $P$ can be obtained from the left subtree, we can prune the right subtree, and vice versa. For both subtrees, the algorithm works correctly by the induction assumption, and thus the logical OR also outputs the correct answer. $\qquad\square$

## C. Additional Material for Section 5

### C.1. Adaption to Reasonable Trees for Theorem 5.1

It remains to show the statement for reasonable trees: First, we extend the classification instance. We add a new binary

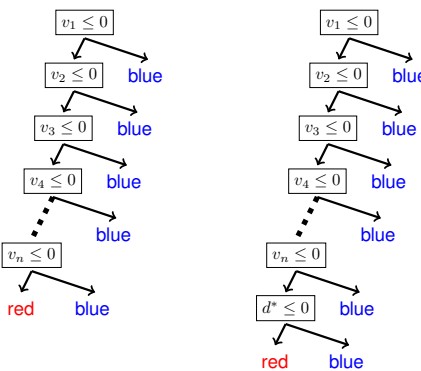

*Figure 7.* Left: The initial decision tree used for Theorems 5.1 to 5.3 and 5.5. Right: The initial reasonable tree used for Theorems 5.1 to 5.3 and 5.5.

feature $d^*$ and we add one new blue example $e_v$ per vertex $v$. Example $e_v$ has value 1 in the feature corresponding to vertex $v$ and in the new feature $d^*$; in all other features (corresponding to any other vertex) $e_v$ has value 0. Also, we add a new blue example $e^*$ for which $e^*[v] = 0$ for all $v \in V(G)$ and $e^*[d^*] = 1$ Furthermore, all existing examples have value 0 in $d^*$.

Second, we extend the tree $T$, that is, we add a new cut $d^* \leq 0$ on the edge leading to the unique red leaf. More precisely, the left child of this node is the unique red leaf and the right child is a blue leaf, see Figure 7. Note that we still have $\delta_{\max} = 2$, $D = 2$, and $t = 0$.

For the correctness, observe that the new cut $d^* \leq 0$ cannot be pruned because of the new blue example $e^*$. The remaining correctness proof is completely analog.

### C.2. Proof of Theorem 5.2

*Proof.* We use the same initial reasonable decision tree as in Theorem 5.1, but instead reduce from $\kappa$-PARTIAL VERTEX COVER (Guo et al., 2007), where we look for a subset of $\kappa$ vertices that covers at least $t'$ edges of $G$. Unless the ETH fails, $\kappa$-PARTIAL VERTEX COVER cannot be solved in $f(\kappa) \cdot n^{o(\kappa)}$ time (Cygan et al., 2015). We let $\ell = \kappa$, $t = |E(G)| - t'$ and create the examples for edges as before. However, we copy the red example $t + 1$ times to prevent pruning the red leaf. Now, if the remaining cuts after pruning correctly classify at least $t'$ blue examples, then at most $t$ blue examples are misclassified. Additionally, we copy the example $e^*$ also $t + 1$ times, to avoid pruning the newly introduced cut $d^* \leq 0$ to make the tree reasonable. Now, the ETH bound follows since $\ell = \kappa$. $\qquad\square$

### C.3. Proof of Theorem 5.3

*Proof.* We use the same initial reasonable decision tree as in Theorem 5.1, but instead reduce from the W[2]-hard prob-

lem $\kappa$-HITTING SET (Cygan et al., 2015), where we look for a subset of $\kappa$ elements from the universe $\mathcal{U}$ that intersects with all subsets given in the input. Unless the ETH fails, $\kappa$-HITTING SET cannot be solved in $f(\kappa) \cdot n^{o(\kappa)}$ time (Cygan et al., 2015). We let $\ell = \kappa$ and $t = 0$. For the examples, we create one blue example $e$ for each subset $S \subseteq \mathcal{U}$ in the input such that $e[u] = 1$ if $u \in S$ and otherwise $e[u] = 0$ for all $u \in \mathcal{U}$. Additionally, we create one red example $e$ with $e[u] = 0$ for all $u \in \mathcal{U}$. Now, the ETH bound follows since $\ell = \kappa$. □

### C.4. Proof of Theorem 5.4

*Proof.* The training data set $(E, \lambda)$ is shown in part $a)$ of Figure 8. More precisely, $(E, \lambda)$ consists of exactly $k$ features and has exactly two red examples and $2 \cdot (k - 1)$ blue examples. The first red example has value 0 is each feature and the second red example has value 0 in each features, except feature $d_1$ where it has value 1. Furthermore, for each $j \in [2, k]$ we have a blue example with value 0 in each feature except feature $d_j$ where the example has value 1 and another blue example with value 0 in each feature except features $d_1$ and $d_j$ where the example has value 1. The initial decision tree is shown in part $b)$ of Figure 8. Note that this tree is reasonable.

On the one hand, if we perform exactly $k$ raising operations, we can prune the root of $T$ and wither its entire left or right subtree. The resulting decision tree has no errors. On the other hand, if we perform $j$ raising operations for some $1 \le j < k$, then we cannot prune the root of $T$. Without loss of generality, we assume that at least one raising operation is done in the left subtree of the root of $T$. Observe that the last cut $d_k \le 0$ cannot be pruned since then a red example ends up in a blue leaf. Similarly, no cut $d_i \le 0$ for some $2 \le i \le k - 1$ cannot be pruned since then a blue example ends up in the red leaf. Consequently, exactly $k$ raising operations are required. □

### C.5. Proof of Theorem 5.5

*Proof.* We use the same initial reasonable decision tree as in Theorem 5.1 and reduce from $\kappa$-INDEPENDENT SET with an instance graph $G$. The blue examples for the edges are constructed similarly to the previous proofs but are duplicated $|V(G)|$ times. We create $|V(G)|^2$ copies of the red example, and finally, construct one red example $e$ for each feature $v$ such that $e[v]$ is 1 and other entries are zeros. By the construction, the initial decision tree has $|V(G)|$ misclassifications.

For the instance of DTRAIS$_\ge$, set $t = |V(G)| - \kappa$. As a consequence, we have to prune at least $\kappa$ cuts to decrease the number of errors to $t$. On the other hand, if cuts corresponding to both endpoints of an edge are pruned, we would create $|V(G)|$ new misclassifications, so the pruned

subset of inner nodes has to be an independent set. Additionally, we copy the example $e^*$ also $t + 1$ times, to avoid pruning the newly introduced cut $d^* \le 0$ to make the tree reasonable. □

### C.6. Proof of Theorem 5.6

*Proof.* We only show he statement for DTRAIS$_=$. The statement for DTRAIS$_\ge$ then follows by setting the lower bound $k$ of the number of pruned inner nodes to the number of inner nodes of the input tree minus $\ell$ (these values are specified later).

We reduce from MULTICOLORED CLIQUE where each color class has the same number $p$ of vertices. Formally, the input is a graph $G$, and $\kappa \in \mathbb{N}$, where the vertex set $V(G)$ of $N$ vertices is partitioned into $V_1, \ldots, V_\kappa$ and $|V_i| = p$ for each $i \in [\kappa]$. More precisely, $V_i := \{v_i^1, v_i^2, \ldots, v_i^p\}$. The question is whether $G$ has contains a clique consisting of exactly one vertex per class $V_i$. MULTICOLORED CLIQUE is W[1]-hard parameterized by $\kappa$ and cannot be solved in $f(\kappa) \cdot n^{o(\kappa)}$ time unless the ETH fails (Cygan et al., 2015).

The property that all color classes have the same number of vertices is only used to simplify the proof.

**Outline:** The idea is to create two features $d_i^<$ and $d_i^>$ per color class $i$ such that in the pruned tree we need to preserve exactly one cut in each feature to fulfill the desired error bound. We achieve the property of exactly one cut per feature $d'$ by adding a huge number of *blue forcing examples* and *red enforcing examples* which can only be distinguished in $d'$. The two cuts in features $d_i^<$ and $d_i^>$ force us to select exactly one vertex $v_i^{a_i}$ of this color class. For each edge we create an *edge example*. If vertex $v_i^{a_i}$ is selected, then all edge examples corresponding to edges having an endpoint in color class $i$ which is *not* $v_i$ will then be misclassified. Thus, we can only correctly classify an edge example if we select both endpoints of that edge. Furthermore, we ensure that only edge examples may be misclassified without violating the error bound. Hence, by setting $t := m - \binom{\kappa}{2}$, we ensure that we need to select a multicolored clique.

**Construction:** We first show the statement for non-reasonable decision trees and afterwards we argue how the construction has to be adapted such that the input decision tree is reasonable.

*Description of the data set:* A visualization is shown in part $b)$ of Figure 9.

- For each edge $\{v_i^a, v_j^b\} \in E(G)$ we add an *edge example* $e(v_i^a, v_j^b)$. To all these examples we assign label red.

- For each $i \in [\kappa]$ and each $a \in [p - 1]$ we add a

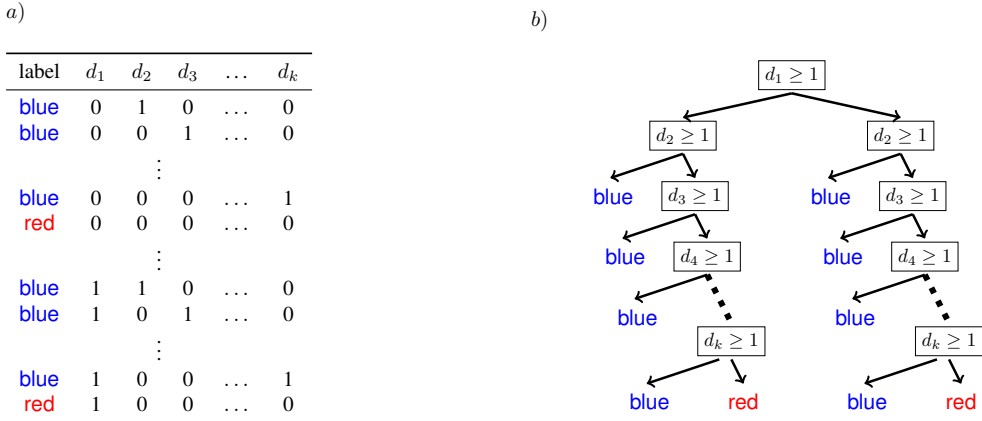

Figure 8. An instance for which pruning fewer than $k$ cuts leads to misclassifications, with examples described by $a)$ and the initial decision tree by $b)$.

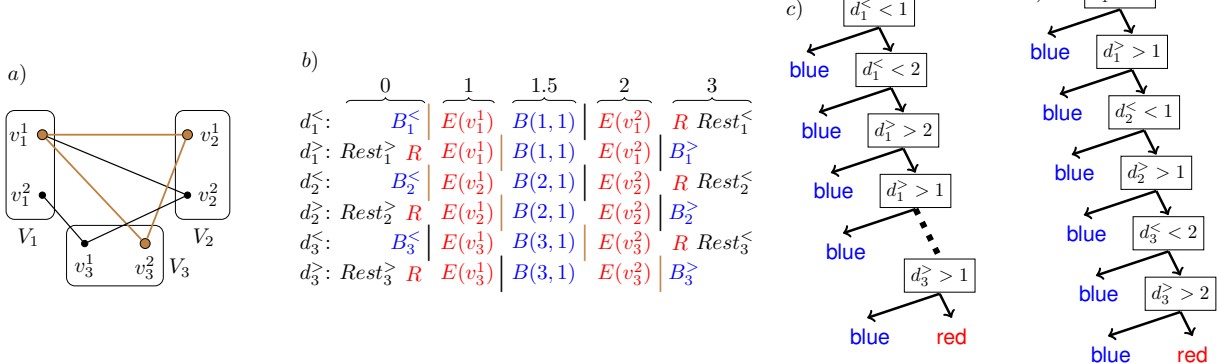

Figure 9. A visualization of the reduction from the proof of Theorem 5.6. $a)$ shows a MULTICOLORED CLIQUE instance. A multicolored clique is depicted in brown. $b)$ shows the corresponding classification instance. Here, $E(v_i^j)$ is the set of all edges incident with vertex $v_i^j$. $Rest_i^<$ and $Rest_i^>$ refers to all other examples not shown in that feature (the precise set differs in each feature and is always a subset of all examples having the default threshold of that feature). Cuts of the input tree $T$ are shown by "$|$" and cuts that remain in the optimal raised tree $T'$ are depicted in brown. $c)$ shows parts of the input tree $T$. $d)$ shows the optimal raised tree $T'$.

set $B(i,a)$ of *separation examples*. Each of these sets consists of $m$ examples having the same value in each feature. To all these examples we assign label blue.

- For each $i \in [\kappa]$ we create sets $B_i^<, B_i^>$ of *blue forcing examples*. Each of these sets consists of $m$ examples and all examples in one of these sets have the same value in each feature.

- We create a set $R$ of *red enforcing examples*. This set consists of $m$ examples and all examples in this set have the same value in each feature.

Note that we add $M := |E(G)|$ edge examples, $\kappa \cdot (p-1) \cdot M \leq N \cdot M$ separation examples, $2 \cdot \kappa \cdot M \leq 2 \cdot N \cdot M$ forcing examples, and $M$ enforcing examples. Thus, the number of examples is polynomial in the input size.

For each $i \in [\kappa]$, we add two features $d_i^<$ and $d_i^>$ and thus

we have $2 \cdot \kappa$ features.

It remains to describe the coordinates of the examples in the features. Initially, we declare a *default threshold* default$(d')$ for each feature $d'$. Then, each example $e$ has the default threshold in each feature, unless we assign $e$ a different threshold in that feature.

For each feature $d_i^<$, we set default$(d_i^<) = p$, and for each feature $d_i^>$, we set default$(d_i^>) = 0$.

- For each edge example $e = e(v_i^a, v_j^b)$ we set $e[d_i^<] = e[d_i^>] = a$, and $e[d_j^<] = e[d_j^>] = b$, $e[d_z^<] = p+1$. In each other features $e$ is set to the default threshold.

- For each separation example $e \in B(i,a)$ we set $e[d_i^<] = e[d_i^>] = a + 1/2$. In each other features $e$ is set to the default threshold.

- For each forcing example $e \in B_i^<$ we set $e[d_i^<] = 0$.

In each other features $e$ is set to the default threshold.

For each forcing example $e \in B_i^>$ we set $e[d_i^>] = p+1$. In each other features $e$ is set to the default threshold.

- For each enforcing example $e \in R$ we use the default threshold in each feature.

*Description of the input tree $T$:* Intuitively, the input tree $T$ is a path, where all leafs are blue except one leaf and we first have some cuts in $d_1^<$ in ascending order, then some cuts in $d_1^>$ in descending order, some cuts in $d_2^<$ in ascending order, and so on. A visualization of $T$ is shown in part $c)$ of Figure 9. Since $T$ is a path it is sufficient to present the order of the cuts from the root to the unique red leaf: $(d_1^< < 1, d_1^< < 2, \ldots, d_1^< < p, d_1^> > p, d_1^> > p-1, \ldots, d_1^> > 1, d_2^< < 1, \ldots, d_\kappa^> > 1)$. The left child of each cut is always a blue leaf and the unique red leaf is the right child of the last cut $d_\kappa^> > 1$. Observe that $T$ consists of $p \cdot \kappa = N$ inner nodes.

*Error bound and $\ell$:* Finally, we set $t := M - \binom{\kappa}{2}$, and $\ell := 2 \cdot \kappa$. This completes our construction.

*Calculation of $\delta_{\max}$:* Observe that each enforcing example always has the default threshold, that each forcing example differs in exactly one features from the default thresholds, that each separation example differs exactly twice from the default thresholds, and that each edge example differs exactly four times from the default thresholds. Thus, $\delta_{\max} = 6$.

**Correctness:** We show that $G$ has a multicolored clique if and only if $T$ can be raised to a tree $T'$ having exactly $\ell$ inner nodes making at most $t = M - \binom{\kappa}{2}$ errors.

$(\Rightarrow)$ Let $S = \{v_i^{a_i} : i \in [\kappa], a_i \in [p]\}$ be a multicolored clique in $G$ (for example: see part $a)$ of Figure 9). To obtain tree $T'$ we preserve the cuts $\{d_i^< < a_i : i \in [\kappa]\}$ and $\{d_i^> > a_i : i \in [\kappa]\}$. A visualization of $T'$ is shown in part $d)$ of Figure 9. In other words, $T'$ is doing the cuts $(d_1^< < a_1, d_1^> > a_1, d_2^< < a_2, \ldots, d_\kappa^> > a_\kappa)$. Clearly, $T'$ consists of $\ell = 2 \cdot \kappa$ inner nodes. Thus, it remains to verify that $T'$ makes at most $t$ errors.

*Outline:* First, we make an observation for examples using the default threshold in a feature and second we use this observation to show that $T'$ has at most $t$ misclassifications.

*Step 1:* Observe that if any example $e$ lands at some inner node of $T$ corresponding to a cut in feature $d'$ and $e$ has the default threshold in that feature $d'$, that is, $e[d'] = \mathsf{default}(d')$, then $e$ will always go to the right subtree of that node. Since the pruned tree $T'$ is a path, an example $e$ which has the default threshold in each feature will be contained in the unique red leaf of $T'$ which is the right leaf of the cut $d_\kappa^> > a_\kappa$. Also, in order for an example $e$ to land in a different leaf (which has label blue), we only need to con-

sider cuts of $T'$ in features where $e$ has a different threshold than the default threshold.

*Step 2:* We distinguish the different example types.

*Step 2.1:* Recall that each enforcing example $e \in R$ always has the default threshold. By Step 1, each forcing example ends up in the unique red leaf and is thus correctly classified in $T'$.

Now, consider a blue forcing example $e \in B_i^<$.

Similar to the enforcing examples, we have $e \in E[d_i^< < a_i]$. By construction, $e$ uses the default thresholds in each feature other than $d_i^<$ and in feature $d_i^<$, we have $e[d_i^<] = 0$. By Step 1 and since $a_i > 0$, $e$ ends up in the left child of the cut $d_i^< < a_i$ which is a blue leaf and thus $e$ is correctly classified in $T'$. Analogously, we can show that all blue forcing examples in $B_i^>$ are correctly classified by $T'$.

Thus, all enforcing and all forcing examples are correctly classified in $T'$.

*Step 2.2:* Let $e \in B(i, b)$ be a blue separation example. By construction, $e$ uses the default thresholds in all features except $d_i^<$ and $d_i^>$. Next, we distinguish the values of $a_i$ and $b$. Recall that $b = q+1/2$ where $q \in \mathbb{N}$ and that $a_i \in [p]$. Thus, either $b < a_i$ or $b > a_i$.

First, consider the case $b < a_i$. Then, $e$ ends up in the left child of the cut $d_i^< < a_i$ which is a blue leaf and thus $e$ is correctly classified.

Second, consider the case $b > a_i$. Then, $e$ ends up in the right child of the cut $d_i^< < a_i$ which is the cut $d_i^> > a_i$. Now, $e$ ends up in the left child of the cut $d_i^> > a_i$ which is a blue leaf and thus $e$ is correctly classified.

Hence, in $T'$ all separation examples are correctly classified.

*Step 2.3:* Let $e = e(v_i^{a_i}, v_j^{a_j})$ be the edge example corresponding to an edge where both endpoints are contained in the multicolored clique $S$. Without loss of generality, we assume $i < j$. Recall that $e$ uses the default thresholds in all features except $d_i^<, d_i^>, d_j^<$, and $d_j^>$. Analogously, to all other example sets, we obtain that $e \in E[d_i^< < a_i]$. Since $e[d_i^<] = a_i$, $e$ ends up in the right child of this node which is the cut $d_i^> > a_i$. Again, since $e[d_i^>] = a_i$, $e$ ends up in the right child of this node. Analogously, we can argue that $e$ always end up in the right child of any cut in $T'$ and thus $e$ ends up in the unique red leaf.

Hence, all edge examples corresponding to edges having both endpoint in the multicolored clique $S$ are correctly classified. Consequently, $T'$ makes at most $t = M - \binom{\kappa}{2}$ errors.

$(\Leftarrow)$ Let $T'$ be a solution for the raising problem, that is, $T'$ has $\ell = 2 \cdot \kappa$ inner nodes and makes at most $t = M - \binom{\kappa}{2}$ errors.

*Outline:* First, we show that we need to preserve exactly one cut per feature. Second, we show that the two cuts in the two features $d_i^<$ and $d_i^>$ need to have the form $d_i^< < a_i$ and $d_i^> > a_i$ for some $a_i \in [p]$. This value $a_i$ then corresponds to vertex $v_i^{a_i}$ of color class $i$. The union of these vertices is $S$. Finally, we verify that $S$ has to be a multicolored clique.

*Step 1:* Assume towards a contradiction that $T'$ does not preserve a cut in each feature, and without loss of generality, assume that no cut in feature $d_i^<$ is preserved in $T'$. Consider the red enforcing examples in $R$ and the blue forcing examples in $B_i^<$. The examples in $R$ always use the default thresholds and the examples in $B_i^<$ use the default thresholds in all features except $d_i^<$. Thus, examples in $R$ and examples in $B_i^<$ can only be distinguished in feature $d_i^<$. Since $T'$ does not preserve a cut in feature $d_i^<$ all examples in $R \cup B_i^<$ end up in the same leaf. Since $|R| = M = |B_i^<|$, we conclude that $T'$ has at least $M > M - \binom{\kappa}{2} = t$ errors, a contradiction. Thus, $T'$ preserves at least one cut per feature.

Since $d = 2 \cdot \kappa = \ell$, we obtain that $T'$ preserves exactly one cut per feature.

*Step 2:* Assume towards a contradiction that the two cuts in features $d_i^<$ and $d_i^>$ do not have the form $d_i^< < a_i$ and $d_i^> > a_i$ for some $a_i \in [p]$, that is, we assume the cuts have the form $d_i^< < a_i$ and $d_i^> > b_i$ with $a_i, b_i \in [p]$ where either $b_i > a_i$ or $b_i < a_i$.

First, we consider the case $b_i > a_i$. We show that all $M$ many blue separation examples in $B(i, a_i)$ end up in the unique red leaf, implying that the number of errors in $T'$ is at least $M > M - \binom{\kappa}{2} = t$, a contradiction. Recall that each example $e \in B(i, a_i)$ uses the default thresholds in all features except $d_i^<$ and $d_i^>$. Thus, in each cut in a feature $d_j^<$ or $d_j^>$ where $j \neq i$, $e$ goes always to the right subtree of that cut. Hence, it remains to consider the cuts in features $d_i^<$ and $d_i^>$. By definition, $e[d_i^<] = a_i + 1/2 = e[d_i^>]$. Since $a_i < a_i + 1/2 < a_i + 1 \leq b_i$, we conclude that $e$ ends up in the right child of both cuts $d_i^< < a_i$ and $d_i^> > b_i$. Consequently, $e$ ends up in the right leaf of the last cut in $T'$ which is red, a contradiction.

Second, we consider the case $b_i < a_i$. Observe that for all examples $e \in E[d_i^> > b_i]$ we have $e[d_i^>] \geq a_i$. More precisely, only for the red edge examples having one endpoint in $v_i^{a_i}$ we have $e[d_i^>] = a_i$ and for all other examples $e' \in E[d_i^> > b_i]$ we have $e'[d_i^>] > a_i$. Since the left child of the cut $d_i^> > b_i$ is a blue leaf, we can replace threshold $b_i$ by threshold $a_i$ without increasing the number of errors. Note that $d_i^> > a_i$ is a cut in the input tree between all cuts in feature $d_i^<$ and $d_{i+1}^<$.

Hence, in the following we can safely assume that for each $i \in [\kappa]$, the cuts in features $d_i^<$ and $d_i^>$ have the form $d_i^< < a_i$ and $d_i^> > a_i$ for some $a_i \in [p]$.

By $v_i^{a_i}$ we denote the vertex which is selected in color class $i$ and by $S$ we denote the set of these vertices.

*Step 3:* We now show that each red edge example corresponding to an edge where at least one vertex is not contained in $S$ ends up in a blue leaf and is thus misclassified. As a consequence, $S$ has to be a multicolored clique to fulfill the error bound of $t = M - \binom{\kappa}{2}$.

Let $e = e(v_i^{a_i}, v_j^{a_j})$ be an edge example and assume without loss of generality that $v_i^{a_i} \notin S$. Recall that $e$ uses the default thresholds in all features except $d_i^<, d_i^>, d_j^<$, and $d_j^>$. Let $v_i^{b_i}$ be the vertex chosen in color class $i$ and assume without loss of generality that $a_i < b_i$. Now, consider the cut in feature $d_i^< < b_i$: since $a_i < b_i$, example $e$ ends up in the left child of the cut $d_i^< < b_i$ which is a blue leaf and is thus misclassified.

**Lower bound:** Recall that $d = 2 \cdot \kappa = \ell$ and $\delta_{\max} = 6$. Since MULTICOLORED CLIQUE is W[1]-hard with respect to $\kappa$ (Cygan et al., 2015), we obtain that DTRAIS$_=$ is W[1]-hard with respect to $d + \ell$ even if $\delta_{\max} = 6$. Furthermore, since MULTICOLORED CLIQUE cannot be solved in $f(\kappa) \cdot n^{o(\kappa)}$ time unless the ETH fails (Cygan et al., 2015), we observe that DTRAIS$_=$ cannot be solved in $f(d + \ell) \cdot |I|^{o(d+\ell)}$ time if the ETH is true, where $|I|$ is the overall instance size, even if $\delta_{\max} = 6$.

**Adaptation for reasonable trees:** We do an analog adaption as in Theorem 5.1: First, we extend the classification instance. Basically, we add one example for each leaf in $T$ which ends up in that specific leaf. For cut $d_i^< < a_i$ we add an example $e$ such that $e[d_i^<] = a_i - 1$, $e[d_z^<] = p + 1$ for each $z \in [\kappa] \setminus \{i\}$, and $e[d_z^>] = 0$ for each $z \in [\kappa]$. For cut $d_i^> > a_i$ we add an example $e$ such that $e[d_z^<] = p + 1$ for each $z \in [\kappa]$, $e[d_i^>] = a_i + 1$, and $e[d_z^>] = 0$ for each $z \in [\kappa] \setminus \{e\}$. Note that all enforcing examples end up in the unique red leaf.

Next, we add a new binary feature $d^*$ and we add $(t+1)$ new blue examples $e^*$ which have the same thresholds in all features. More precisely, $e^*$ has threshold 1 in $d^*$ and uses the default threshold in each remaining feature. All other existing examples have threshold 0 in the new feature $d^*$. Example $e_v$ has value 1 in the feature corresponding to vertex $v$ and in the new feature $d^*$; in all other features (corresponding to any other vertex) $e_v$ has value 0. Also, we add a new blue example $e^*$ for which $e^*[v] = 0$ for all $v \in V(G)$ and $e^*[d^*] = 1$ Furthermore, all existing examples have value 0 in $d^*$.

Now, observe that the newly added cut $d^* \leq 0$ cannot be pruned since otherwise the $(t + 1)$ newly added examples would be misclassified. Afterwards, the correctness can be shown analogously. Note that this adaption increased $d$

and $\ell$ by one and does not change $\delta_{\max}$. $\qquad \square$

### C.7. Proof of Proposition 5.7

*Proof.* The proof is almost identical to the proof of Theorem 5.6 for both non-reasonable and reasonable trees. More precisely, we use the same construction. Moreover, the $(\Rightarrow)$ direction of the correctness works analogously. The $(\Leftarrow)$ direction of the correctness is shown with the same three steps. Now, however, Step 1, that is, exactly one cut per feature is preserved, is more involved since we cannot exploit anymore that exactly $2 \cdot \kappa$ cuts are preserved. After we have verified Step 1, the remaining proof works analogously. Thus, it remains to show that exactly one cut per feature is preserved.

Assume that in the resulting tree $T'$ at least two cuts in one feature are preserved. Without loss of generality assume that this is the case in feature $d_i^<$, that is, $T'$ contains two cuts $d_i^< < a$ and $d_i^< < b$ for some $a < b$. Furthermore, assume without loss of generality that no other cut in $d_i^<$ between $a$ and $b$ is preserved. Observe that all examples $e$ with $e[d_i^<] < a$ end up in a blue leaf and also all examples $e'$ with $a \le d_i^<[e'] < b$ end up in a blue leaf. Since $d_i^< < a$ is the parent of $d_i^< < b$ in $T'$, raising $d_i^< < a$ leads to a smaller tree $T''$ which misclassifies the exact same set of examples. By applying this argument iteratively, we obtain a tree $T^*$ with exactly one cut in each feature and thus Step 1 is verified. $\qquad \square$

### C.8. Proof of Theorem 5.8

*Proof.* We only show the statement for DTRAIS$_=$. The statement for DTRAIS$_\ge$ then follows since no more than $k$ inner nodes can be pruned without having at least 1 error.

We reduce from MULTICOLORED INDEPENDENT SET where each color class has the same number $p$ of vertices. Formally, the input is a graph $G$, and $\kappa \in \mathbb{N}$, where the vertex set $V(G)$ of $N$ vertices is partitioned into $V_1, \ldots, V_\kappa$ and $|V_i| = p$ for each $i \in [\kappa]$. More precisely, $V_i := \{v_i^1, v_i^2, \ldots, v_i^p\}$ and $p \cdot \kappa = N$. The question is whether $G$ contains an independent set consisting of exactly one vertex per class $V_i$. MULTICOLORED INDEPENDENT SET is W[1]-hard parameterized by $\kappa$ and cannot be solved in $f(\kappa) \cdot n^{o(\kappa)}$ time unless the ETH fails (Cygan et al., 2015).

As in the proof of Theorem 5.6, the property that all color classes have the same number of vertices is only used to simplify the proof.

**Outline:** The idea is to create two features $d_i^<$ and $d_i^>$ per color class $i$ such that the preserved cuts in the pruned tree $T'$ correspond to a vertex selection in $V_i$. We achieve this as follows: For each pair $d_i^<$ and $d_i^>$ of features we create examples which can only be separated in these two features and which have labels blue (*separating examples*) and red (*choice examples*) alternatingly. Hence, for each possible threshold $x$ in features $d_i^<$ and $d_i^>$, we either need to preserve cut $(d_i^<, x)$ or cut $(d_i^>, x)$. Furthermore, for each edge we create a red *edge example*. If vertex $v_i^j \in V_i$ is selected, then all edge examples corresponding to edges having an endpoint in color class $i$ which is *not* $v_i$ will then be correctly classified by the pruned tree $T'$. Thus, we can only correctly classify an edge example if we *do not* select at least one endpoint of the corresponding edge. Finally, we have another feature $d^*$ with only 2 thresholds to ensure that all red choice examples corresponding to selected vertices are correctly classified by the pruned tree $T'$ and that an edge example gets misclassified as blue if we select both endpoints of the corresponding edge.

**Construction:** We first show the statement for non-reasonable decision trees and afterwards we argue how the construction has to be adapted such that the input decision tree is reasonable.

*Description of the data set:* A visualization is shown in part $b)$ of Figure 10.

- For each edge $\{v_i^x, v_j^z\} \in E(G)$ we add an *edge example* $e(v_i^x, v_j^z)$. To all these examples we assign label red.

- For each $i \in [\kappa]$ and each $x \in [p-1]$ we add a blue *separating example* $b(i, x)$.

- For vertex $v_i^x \in V_i$ we create a red *choice example* $c_i^x$.

- we create a blue *forcing example* $b^*$ and a red *enforcing example* $r^*$.

Note that we add $M := |E(G)|$ edge examples, $N$ choice examples, $N - \kappa = (p-1) \cdot \kappa$ separating examples, and 2 further examples. Thus, the number of examples is polynomial in the input size.

For each $i \in [\kappa]$, we add two features $d_i^<$ and $d_i^>$. We also add another feature $d^*$. Thus, we have $2 \cdot \kappa + 1$ features.

It remains to describe the coordinates of the examples in the features. Initially, we declare a *default threshold* default($d'$) for each feature $d'$. Then, each example $e$ has the default threshold in each feature, unless we assign $e$ a different threshold in that feature.

For each feature $d_i^<$, we set default($d_i^<$) $= p$, for each feature $d_i^>$, we set default($d_i^>$) $= 1$, and for feature $d^*$, we set default($d^*$) $= 2$.

- For each edge example $e = e(v_i^x, v_j^z)$ we set $e[d_i^<] = e[d_i^>] = x$, $e[d_j^<] = e[d_j^>] = z$, and $e[d^*] = 1$. In each other features $e$ is set to the default threshold.

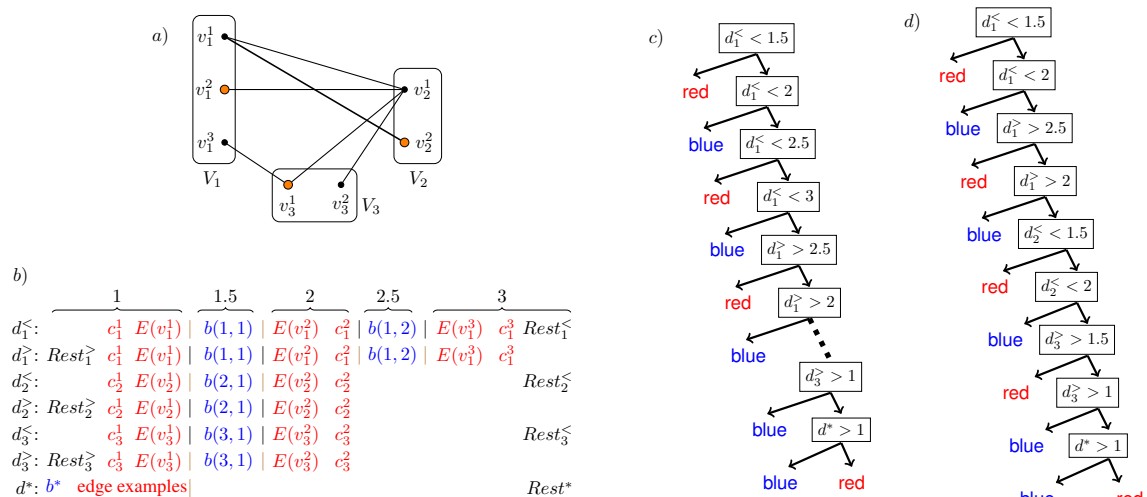

**Figure 10.** A visualization of the reduction from the proof of Theorem 5.8. $a$) shows a MULTICOLORED INDEPENDENT SET instance. For the sake of the illustration, the property that all partite sets have the same size is dropped. A multicolored independent set is depicted in orange. $b$) shows the corresponding classification instance. Here, $E(v_i^j)$ is the set of all edges incident with vertex $v_i^j$. $Rest_i^<$, $Rest_i^>$, and $Rest^*$ refers to all other examples not shown in that feature (the precise set differs in each feature and is always a subset of all examples having the default threshold of that feature). Cuts of the input tree $T$ are shown by "|" and cuts that remain in the optimal raised tree $T'$ are depicted in brown. $c$) shows parts of the input tree $T$. $d$) shows the optimal raised tree $T'$.

- For the separating example $e = b(i, x)$ we set $e[d_i^<] = e[d_i^>] = x + 1/2$. In each other features $e$ is set to the default threshold.

- For the choice example $e = c_i^x$ we set $e[d_i^<] = e[d_i^>] = x$. In each other features $e$ is set to the default threshold.

- The red enforcing example $r^*$ has the default threshold in every feature. For the blue forcing example $b^*$, we set $b^*[d^*] = 1$, and in each other feature we use the default threshold.

Note that each feature has at most $2p-1$ different thresholds.

*Description of the input tree $T$:* Intuitively, the input tree $T$ is a path and we first have the cuts in $d_1^<$ in ascending order, then the cuts in $d_1^>$ in descending order, the cuts in $d_2^<$ in ascending order, and so on, until the cut $d^* > 1$. A visualization of $T$ is shown in part $c$) of Figure 10. Since $T$ is a path it is sufficient to present the order of the cuts starting at the root: $(d_1^< < 1.5, d_1^< < 2, d_1^< < 2.5, \ldots, d_1^< < p, d_1^> > p - 1/2, d_1^> > p - 1, \ldots, d_1^> > 1, d_2^< < 1.5, \ldots, d_\kappa^> > 1, d^* > 1)$. Let $x \in [p - 1]$. For each $i \in [\kappa]$, the left child of the cut $d_i^< < x + 1/2$ and the left child of the cut $d_i^> > x + 1/2$ is a red leaf. Also, the right child of the cut $d^* > 1$ is a red leaf. All remaining leaves are blue.

Observe that $T$ consists of $(p - 1) \cdot 4\kappa + 1$ inner nodes.

*Error bound and $\ell$:* Finally, we set $t := 0$, and $\ell := (p - $

$1) \cdot 2\kappa + 1$. Thus, $k = (p - 1) \cdot 2\kappa$. This completes our construction.

*Calculation of $\delta_{\max}$:* Note that each edge example differs at most 4 times from the default thresholds, that each separating and each choice example differs exactly 2 times from the default thresholds, that $b^*$ differs exactly one from the default thresholds, and that $r^*$ always has the default thresholds. Thus, $\delta_{\max} = 6$.

**Correctness:** We show that $G$ has a multicolored independent set if and only if $T$ can be raised to a tree $T'$ having exactly $\ell$ inner nodes making at most $t = 0$ errors.

($\Rightarrow$) Let $S = \{v_i^{a_i} : i \in [\kappa], a_i \in [p]\}$ be a multicolored independent set in $G$ (for example: see part $a$) of Figure 10). In the pruned tree $T'$, for each feature $d_i^<$, we preserve all cuts at thresholds $x \in \{1.5, 2, 2.5, 3, \ldots, p\}$ for which $x \le a_i$. Similarly, for each feature $d_i^>$, we preserve all cuts at thresholds $x \in \{1, 1.5, 2, 2.5, \ldots, p - 1/2\}$ for which $a_i \le x$. Further, we preserve the unique cut in feature $d^*$. In other words, in $T'$ the cuts $\{d_1^< < 1.5, d_1^< < 2, \ldots, d_1^< < a_1, d_1^> > p - 1/2, \ldots, d_1^> > a_1, \ldots, d_\kappa^> > a_\kappa, d^* > 1\}$ are preserved in that specific order. A visualization of $T'$ is shown in part $d$) of Figure 10. Clearly, $T'$ consists of $\ell = (p - 1) \cdot 2\kappa + 1$ inner nodes. Thus, it remains to verify that $T'$ makes no errors.

*Outline:* First, we make an observation for examples using the default threshold in a feature and second we use this observation to show that all examples are correctly classified by the pruned tree $T'$.

*Step 1:* Observe that if any example $e$ lands at some inner node of $T$ corresponding to a cut in feature $d'$ and $e$ has the default threshold in that feature $d'$, that is, $e[d'] = \mathsf{default}(d')$, then $e$ will always go to the right subtree of that node. Since the pruned tree $T'$ is a path, an example $e$ which has the default threshold in each feature will be contained in the red leaf of the cut $d^* > 1$. Also, in order for an example $e$ to land in a different leaf, we only need to consider cuts of $T'$ in features where $e$ has a different threshold than the default threshold.

*Step 2:* We distinguish the different example types.

*Step 2.1:* By construction, the red enforcing example $r^*$ always has the default threshold. Thus $r^*$ ends up in the right child of the last cut of $T'$ which is a red leaf. Furthermore, the unique feature in which the blue forcing example $b^*$ does not have the default threshold is $d^*$. Thus, $b^*$ ends up in the left child of the last cut of $T'$ which is a blue leaf.

Thus, examples $r^*$ and $b^*$ are correctly classified by $T'$.

*Step 2.2:* Consider a blue separating example $e = b(i, z)$. Recall that $z = x + 1/2$ and $x \in [p-1]$ and recall that $a_i \in \mathbb{N}$ is the index of the selected vertex of color class $i$. Without loss of generality, assume that $z < a_i$. By Step 1, $e$ will end up in the cut $d_i^< < 1.5$ of $T'$. Also, recall that the next cuts in $T'$ are $d_i^< < 2, \dots, d_i^< < a_i$ in that specific order. Consequently, $e$ goes to the left subtree of the cut $d_i^< < z + 1/2$, which by construction is a blue leaf. Thus, $e$ is correctly classified as blue by $T'$.

*Step 2.3:* Consider a red choice example $e = c_i^x$ where $x \in [p]$.

First, consider the case that $x \neq a_i$. Then the argumentation is almost identical to the blue separating examples: Without loss of generality, assume that $x < a_i$. By Step 1, $e$ will end up in the cut $d_i^< < 1.5$ of $T'$. Also, recall that the next cuts in $T'$ are $d_i^< < 2, \dots, d_i^< < a_i$ in that specific order. Consequently, $e$ goes to the left subtree of the cut $d_i^< < x + 1/2$, which by construction is a red leaf.

Second, consider the case that $x = a_i$. Observe that in all cuts of $T'$ in features $d_i^<$ and $d_i^>$, example $e$ will always go to the right subtree. Since $e$ has the default threshold in each features different from $d_i^<$ and $d_i^>$, example $e$ ends up in the right leaf of the last cut $d^* > 1$ of $T'$ which is a red leaf.

Thus, in both cases $e$ is correctly classified as red by $T'$.

*Step 2.4:* Consider a red edge example $e = (v_i^x, v_j^z)$. By assumption, $S$ is a multicolored independent set. Hence, at least one of the two endpoints $v_i^x$ and $v_j^z$ is not contained in $S$. Without loss of generality, assume that $v_i^x \notin S$ and that $i < j$. The argumentation is analog to the red choice examples $c_i^x$ where $x \neq a_i$: Without loss of generality, assume that $x < a_i$. By Step 1, $e$ will end up in the cut $d_i^< <$

1.5 of $T'$. Also, recall that the next cuts in $T'$ are $d_i^< < 2, \dots, d_i^< < a_i$ in that specific order. Consequently, $e$ goes to the left subtree of the cut $d_i^< < x + 1/2$, which by construction is a red leaf. Thus, $e$ is correctly classified as red by $T'$.

Consequently, the raised tree $T'$ has no classification errors.

($\Leftarrow$) Let $T'$ be a solution for the raising problem, that is, $T'$ has $\ell = (p-1) \cdot 2\kappa + 1$ inner nodes and has no classification errors.

*Outline:* We first show that the unique cut in feature $d^*$ has to be preserved. Second, we show that at least on of the two cuts $d_i^< < x$ and $d_i^> > x - 1/2$ for each $i$ and each $x$ has to be preserved in $T'$. Third, because of our choice of $\ell$ we then conclude that for each $i$ and each $x$ exactly one of the cuts $d_i^< < x$ and $d_i^> > x - 1/2$ has to be preserved. Fourth, we show that cuts preserved in a feature $d_i^<$ (or $d_i^>$) do not have *gaps*, that is, if $x$ is the largest (smallest) threshold, such that the cut $d_i^< < x$ ($d_i^> > x$) is preserved in $T'$, then also all cuts $d_i^< < z$ for each $z < x$ ($d_i^> > z$ for each $x < z$) have to be preserved in $T'$. Fifth, we use this solution structure to identify a selected vertex of each color class. Let $S$ be the corresponding vertex set. Finally, we show that $S$ has to be a multicolored independent set.

*Step 1:* Note that the blue forcing example $b^*$ and that the red enforcing example $r^*$ only differ in feature $d^*$. Since $T$ has only one cut in feature $d^*$, in the solution $T'$ the cut $d^* > 1$ has to be preserved.

*Step 2:* Our aim is to show that at least one of the cuts $d_i^< < x$ and $d_i^> > x - 1/2$ for any $i \in [\kappa]$ and $x \in \{1.5, 2, 2.5, \dots, p\}$ has to be preserved in $T'$. Without loss of generality assume that $x$ is an integer. Note that $x \geq 2$. By construction, for the blue separating example $e = b(i, x-1)$ we have $e[d_i^<] = e[d_i^>] = x - 1/2$ and for the red choice example $e = c_i^x$ we have $e[d_i^<] = e[d_i^>] = x$. Furthermore, note that $b(i, x - 1)$ and $c_i^x$ have the default threshold in each other feature. Consequently, only the cuts $d_i^< < x$ and $d_i^> > x - 1/2$ of $T$ can separate $b(i, x-1)$ and $c_i^x$. Since $T'$ has no classification errors, we thus conclude that at least one of the cuts $d_i^< < x$ and $d_i^> > x - 1/2$ has to be preserved in $T'$.

*Step 3:* Recall that $\ell = (p - 1) \cdot 2\kappa + 1$. By Step 1, in $T'$ the cut $d^* > 1$ has to be preserved. By Step 2, at least one of the cuts $d_i^< < x$ and $d_i^> > x - 1/2$ for any $i \in [\kappa]$ and $x \in \{1.5, 2, 2.5, \dots, p\}$ has to be preserved in $T'$. Note that these are exactly $(p - 1) \cdot 2\kappa$ pairs of distinct cuts. Consequently, in $T'$ exactly one of the cuts $d_i^< < x$ and $d_i^> > x - 1/2$ has to be preserved.

*Step 4:* Consider all preserved cuts in feature $d_i^<$. Let $d_i^< < x$ be the rightmost preserved cut in feature $d_i^<$, that is, for each $z > x$, the cut $d_i^< < z$ is pruned. We claim that in $T'$

all cuts $d_i^< < z$ for any $z \leq x$ have to be preserved. Without loss of generality, we assume that $x$ is an integer. Since all blue separating examples and red choice examples with threshold $w \leq x$ in feature $d_i^<$ have the default feature in all features except $d_i^<$ and $d_i^>$ and since all cuts in feature $d_i^<$ appear before the cuts in feature $d_i^>$ in $T$ and thus also in $T'$, we conclude that all these examples end up in a left leaf of one of the cuts $d_i^< < z$ for some $z \leq x$. Now, observe that in feature $d_i^<$ the examples $c_i^1, b(i, 1), c_i^2, \ldots, c_i^{x-1}, b(i, x-1)$ have strictly monotone increasing thresholds and have alternating labels red and blue. Consequently, all cuts of the form $d_i^< < z$ for each $z \leq x$ need to be preserved in $T'$.

By the above argumentation and Step 3, we conclude that in feature $d_i^>$ all cuts of the from $d_i^> > z$ for each $z > x$ are preserved in $T'$.

*Step 5:* Consider one fixed $i \in [\kappa]$. Let $x_i$ be the largest threshold such that the cut $d_i^< < x_i$ is preserved in $T'$. Recall that according to Step 4 all cuts $d_i^< < z$ for any $z \leq x_i$ are preserved in $T'$. Assume towards a contradiction that $x$ is no integer, that is, $x_i = q + 1/2$ for some integer $q \in [p-1]$. Now, observe that the blue separating example $b(i, q)$ is put in the right subtree of each cut $d_i^< < z$ for each $z \leq x_i$ and for each cut $d_i^> > z$ for each $x_i \leq z$. Also, since $b(i, q)$ has the default threshold in each other feature, we conclude that in $T'$ this example $b(i, q)$ ends up in the right leaf of the last cut $d^* > 1$ of $T'$ which is a red leaf. Thus, $b(i, q)$ is misclassified, a contradiction. Hence, $x_i$ is an integer.

We let $v_i^{x_i}$ be the selected vertex of color class $i$. Furthermore, let $S \coloneqq \{v_i^{x_i} : i \in [\kappa]\}$.

*Step 6:* It remains to verify that $S$ is a multicolored independent set. By definition, $S$ contains exactly one vertex of each color class. Hence, it remains to show that $S$ is an independent set.

Observe that since $T'$ has no classification errors, it is sufficient to show that an edge example $e = (v_i^x, v_j^z)$ gets misclassified by $T'$ if both endpoints $v_i^x$ and $v_j^z$ are contained in $S$.

Let $e = e(v_i^x, v_j^z)$ be an edge example where $v_i^x, v_j^z \in S$. Note that analog to Step 5, the red edge example $e$ ends up in the right subtree of each preserved cut in features $d_i^<, d_i^>$, $d_j^<$, and $d_j^>$. The same is true for each feature $d_q^<$ and $d_q^>$ for each $q \in [\kappa] \setminus \{i, j\}$ since in these features $e$ has the default threshold. But in feature $d^*$ example $e$ has not the default threshold, and thus $e$ ends up in the left leaf of the cut $d^* < 1$ which is a blue leaf, a contradiction.

**Lower bound:** Recall that $d = 2 \cdot \kappa + 1$, $\delta_{\max} = 6$, and $t = 0$. Since MULTICOLORED INDEPENDENT SET is W[1]-hard with respect to $\kappa$ (Cygan et al., 2015), we obtain that DTRAIS$_=$ is W[1]-hard with respect to $d$ even if $\delta_{\max} = 6$

and $t = 0$. Furthermore, since MULTICOLORED INDEPENDENT SET cannot be solved in $f(\kappa) \cdot n^{o(\kappa)}$ time unless the ETH fails (Cygan et al., 2015), we observe that DTRAIS$_=$ cannot be solved in $f(d) \cdot |I|^{o(d)}$ time if the ETH is true, where $|I|$ is the overall instance size, even if $\delta_{\max} = 6$ and $t = 0$.

**Adaption for reasonable trees.** Note that some leafs corresponding to cuts in feature $d_i^>$ are not reached by any example. To make the tree $T$ reasonable we do a similar adaption as in Theorem 5.1: For each leaf $L$ of a cut in feature $d_i^<$ (or $d_i^>$) we add a new example which $e$ which uses the default threshold in all features except (a) in $d_i^<$ (or $d_i^>$) and (b) in $d^*$, where $e[d^*] = 0$, if leaf $L$ is blue and otherwise if $L$ is red, then $e[d^*] = 1$. Observe that the correctness can be shown analogously to the non-reasonable case. $\square$

# D. Additional Material for Section 6

Our empirical study aimed to assess whether common decision-tree pruning heuristics achieve optimal tradeoffs between pruned nodes and classification errors. This is made feasible using the algorithmic and complexity analysis in the preceding sections. To this end, we selected benchmark instances used for computing minimum-size trees, as they would likely be suitable for exact algorithms for pruning trees as well. We computed unpruned and pruned trees using the WEKA library. We then took the unpruned trees and computed the whole Pareto front that contains for each number $k$ of pruned nodes, the minimum-possible classification error of the resulting pruned tree. Then we compared the Pareto front to the trade-offs chosen by the heuristics.

We used 40 datasets from the Penn Machine Learning Benchmarks library (Romano et al., 2022). 35 of the datasets were used before for computing minimum-size trees (Bessiere et al., 2009; Narodytska et al., 2018) and since the number of examples was usually small we added further larger datasets. Overall, the datasets range from 72 to 5404 examples (mean 674.88, median 302); for the full details, see Table 2. To meet the requirements of DTRAIS$_=$ inputs, we transformed the data sets as follows (similarly to Janota & Morgado (2020)): First, we replaced each categorical feature by a set of new binary features indicating whether an example is in the category. Second, we converted each instance into a binary classification problem by making two classes, one of which contains all examples of the largest original class and one which contains all remaining examples. Finally, if two examples of different classes had the same value in all features, we removed one of them arbitrarily.

We computed unpruned and pruned trees using the C4.5 heuristic for decision-tree computation (Quinlan, 1993)

implemented WEKA 3.8.5 (Frank et al., 2010). The unpruned trees were obtained by running WEKA's J48 classifier with the flags `-no-cv -B -M 0 -J -U`. We obtained two types of pruned trees: Those obtained by the *replacement heuristic* implemented in J48 when run with the flags `-no-cv -B -M 0 -J -S` and those obtained by the *raising heuristic*, corresponding to the flags `-no-cv -B -M 0 -J`. Overall, the tree size $s$ ranges from 3 to 607 (mean 60.57, median 26); the number of features $d$ ranges from 2 to 88 (mean 12.00, median 9); the domain size $D$ ranges from 1 to 321 (mean 15.55, median 6); the maximum number of features $d_R$ on a root-to-leaf path ranges from 2 to 34 (mean 7.88, median 8); the number of classification errors ranges from 0 to 302 (mean 25.17, median 4). These ranges show that, indeed, parameters $d, D, d_T$ are suitable for designing fixed-parameter algorithms. For the full details, see Table 3.

We implemented a dynamic-programming algorithm for solving DTR$_{\text{AIS}=}$ based on Theorem 4.2 in Python, tested with versions 3.6.9 and 3.10.12. We ran the implementation under Ubuntu Linux 18.04 on a compute cluster with Intel Xeon E5-2640 processors, setting a maximum RAM limit of 64GB. We ran the algorithm for each dataset together with its unpruned tree to obtain, for each number $k$ of pruned nodes, the least number of classification errors. After at most 24h of running time, 26 of the 40 datasets (65 %) could be solved.

*Table 2.* Dataset statistics.

| Dataset | # Examples $n$ | # Features $d$ | Class 0 | Class 1 | Class Ratio |
|---|---|---|---|---|---|
| appendicitis | 106 | 7 | 85 | 21 | 0.25 |
| australian | 690 | 18 | 383 | 307 | 0.8 |
| backache | 180 | 55 | 155 | 25 | 0.16 |
| banana | 5300 | 2 | 2924 | 2376 | 0.81 |
| biomed | 209 | 14 | 75 | 134 | 1.79 |
| breast-cancer | 266 | 31 | 188 | 78 | 0.41 |
| bupa | 341 | 5 | 168 | 173 | 1.03 |
| cars | 392 | 12 | 147 | 245 | 1.67 |
| cleve | 302 | 27 | 164 | 138 | 0.84 |
| cleveland | 303 | 27 | 139 | 164 | 1.18 |
| cleveland-nominal | 130 | 17 | 61 | 69 | 1.13 |
| colic | 357 | 75 | 134 | 223 | 1.66 |
| contraceptive | 1358 | 21 | 764 | 594 | 0.78 |
| dermatology | 366 | 129 | 254 | 112 | 0.44 |
| diabetes | 768 | 8 | 268 | 500 | 1.87 |
| ecoli | 327 | 7 | 184 | 143 | 0.78 |
| flare | 1066 | 10 | 884 | 182 | 0.21 |
| glass | 204 | 9 | 128 | 76 | 0.59 |
| glass2 | 162 | 9 | 86 | 76 | 0.88 |
| haberman | 283 | 3 | 73 | 210 | 2.88 |
| hayes-roth | 84 | 15 | 59 | 25 | 0.42 |
| heart-c | 302 | 27 | 138 | 164 | 1.19 |
| heart-h | 293 | 29 | 106 | 187 | 1.76 |
| heart-statlog | 270 | 25 | 150 | 120 | 0.8 |
| hepatitis | 155 | 39 | 123 | 32 | 0.26 |
| Hill_Valley_without_noise | 1212 | 100 | 600 | 612 | 1.02 |
| hungarian | 293 | 29 | 187 | 106 | 0.57 |
| ionosphere | 351 | 34 | 126 | 225 | 1.79 |
| lupus | 86 | 3 | 52 | 34 | 0.65 |
| lymphography | 148 | 50 | 67 | 81 | 1.21 |
| molecular_biology_promoters | 106 | 228 | 53 | 53 | 1.0 |
| new-thyroid | 215 | 5 | 65 | 150 | 2.31 |
| phoneme | 5404 | 5 | 3818 | 1586 | 0.42 |
| pima | 768 | 8 | 500 | 268 | 0.54 |
| postoperative-patient-data | 72 | 22 | 50 | 22 | 0.44 |
| schizo | 340 | 14 | 140 | 200 | 1.43 |
| soybean | 622 | 133 | 545 | 77 | 0.14 |
| tae | 106 | 5 | 71 | 35 | 0.49 |
| titanic | 2099 | 8 | 1418 | 681 | 0.48 |
| tokyo1 | 959 | 44 | 346 | 613 | 1.77 |

| Dataset | Size $s$ | Dimensions $d$ | Dim. $d_R$ on Path | Domain $D$ | Errors | Error Ratio (%) |
|---|---|---|---|---|---|---|
| appendicitis | 15 / 10 / 10 | 6 / 6 / 6 | 5 / 5 / 5 | 5 / 3 / 3 | 0 / 2 / 2 | 0.00 / 1.89 / 1.89 |
| australian | 90 / 46 / 44 | 13 / 11 / 11 | 10 / 10 / 9 | 29 / 11 / 11 | 0 / 22 / 23 | 0.00 / 3.19 / 3.33 |
| backache | 26 / 13 / 13 | 13 / 9 / 9 | 9 / 6 / 6 | 7 / 2 / 2 | 0 / 7 / 7 | 0.00 / 3.89 / 3.89 |
| banana | 607 / 186 / 188 | 2 / 2 / 2 | 2 / 2 / 2 | 321 / 107 / 108 | 1 / 249 / 246 | 0.02 / 4.70 / 4.64 |
| biomed | 21 / 3 / 13 | 6 / 3 / 6 | 6 / 3 / 6 | 6 / 1 / 4 | 0 / 15 / 3 | 0.00 / 7.18 / 1.44 |
| breast-cancer | 95 / 31 / 24 | 25 / 21 / 17 | 14 / 10 / 9 | 8 / 6 / 5 | 2 / 31 / 33 | 0.75 / 11.65 / 12.41 |
| bupa | 111 / 72 / 65 | 5 / 5 / 5 | 5 / 5 / 5 | 21 / 18 / 14 | 0 / 25 / 28 | 0.00 / 7.33 / 8.21 |
| cars | 22 / 15 / 15 | 7 / 7 / 7 | 5 / 5 / 5 | 7 / 5 / 5 | 0 / 3 / 3 | 0.00 / 0.77 / 0.77 |
| cleve | 57 / 29 / 29 | 15 / 13 / 13 | 9 / 8 / 8 | 12 / 5 / 5 | 0 / 15 / 15 | 0.00 / 4.97 / 4.97 |
| cleveland | 55 / 31 / 31 | 16 / 13 / 13 | 8 / 8 / 8 | 10 / 6 / 6 | 0 / 13 / 13 | 0.00 / 4.29 / 4.29 |
| cleveland-no... | 46 / 8 / 8 | 15 / 8 / 8 | 10 / 5 / 5 | 1 / 1 / 1 | 6 / 23 / 23 | 4.62 / 17.69 / 17.69 |
| colic | 51 / 28 / 28 | 27 / 18 / 18 | 9 / 8 / 8 | 6 / 4 / 4 | 0 / 15 / 15 | 0.00 / 4.20 / 4.20 |
| contraceptive | 486 / 120 / 106 | 21 / 21 / 21 | 13 / 11 / 11 | 33 / 21 / 21 | 10 / 217 / 224 | 0.74 / 15.98 / 16.49 |
| dermatology | 5 / 3 / 3 | 4 / 3 / 3 | 3 / 3 / 3 | 1 / 1 / 1 | 0 / 2 / 2 | 0.00 / 0.55 / 0.55 |
| diabetes | 137 / 96 / 87 | 8 / 8 / 8 | 8 / 8 / 8 | 24 / 16 / 14 | 0 / 24 / 30 | 0.00 / 3.12 / 3.91 |
| ecoli | 25 / 5 / 5 | 5 / 3 / 3 | 4 / 3 / 3 | 11 / 2 / 2 | 0 / 10 / 10 | 0.00 / 3.06 / 3.06 |
| flare | 93 / 15 / 12 | 8 / 7 / 6 | 8 / 7 / 6 | 5 / 4 / 2 | 125 / 159 / 160 | 11.73 / 14.92 / 15.01 |
| glass | 28 / 26 / 24 | 7 / 7 / 7 | 7 / 7 / 7 | 7 / 7 / 6 | 0 / 1 / 2 | 0.00 / 0.49 / 0.98 |
| glass2 | 22 / 16 / 14 | 6 / 5 / 5 | 5 / 4 / 4 | 6 / 5 / 4 | 0 / 4 / 5 | 0.00 / 2.47 / 3.09 |
| haberman | 92 / 21 / 18 | 3 / 3 / 3 | 3 / 3 / 3 | 30 / 9 / 8 | 2 / 38 / 39 | 0.71 / 13.43 / 13.78 |
| hayes-roth | 14 / 12 / 12 | 11 / 10 / 10 | 10 / 10 / 10 | 1 / 1 / 1 | 0 / 1 / 1 | 0.00 / 1.19 / 1.19 |
| heart-c | 57 / 29 / 29 | 15 / 13 / 13 | 9 / 8 / 8 | 12 / 5 / 5 | 0 / 15 / 15 | 0.00 / 4.97 / 4.97 |
| heart-h | 57 / 32 / 30 | 20 / 18 / 18 | 13 / 11 / 10 | 13 / 6 / 6 | 0 / 14 / 14 | 0.00 / 4.78 / 4.78 |
| heart-statlog | 54 / 27 / 23 | 17 / 13 / 12 | 10 / 10 / 9 | 13 / 7 / 6 | 0 / 15 / 17 | 0.00 / 5.56 / 6.30 |
| hepatitis | 18 / 12 / 11 | 10 / 9 / 8 | 7 / 7 / 6 | 3 / 2 / 2 | 0 / 3 / 4 | 0.00 / 1.94 / 2.58 |
| Hill_Valley... | 250 / 228 / 224 | 88 / 85 / 84 | 34 / 34 / 33 | 63 / 47 / 47 | 0 / 11 / 13 | 0.00 / 0.91 / 1.07 |
| hungarian | 57 / 32 / 30 | 19 / 19 / 18 | 13 / 11 / 10 | 13 / 6 / 6 | 0 / 14 / 14 | 0.00 / 4.78 / 4.78 |
| ionosphere | 21 / 19 / 19 | 12 / 11 / 11 | 9 / 9 / 9 | 4 / 4 / 4 | 0 / 1 / 1 | 0.00 / 0.28 / 0.28 |
| lupus | 25 / 4 / 4 | 2 / 2 / 2 | 2 / 2 / 2 | 20 / 2 / 2 | 0 / 13 / 13 | 0.00 / 15.12 / 15.12 |
| lymphography | 23 / 14 / 14 | 18 / 11 / 11 | 10 / 6 / 6 | 1 / 1 / 1 | 0 / 5 / 5 | 0.00 / 3.38 / 3.38 |
| molecular_b... | 12 / 10 / 10 | 11 / 9 / 9 | 6 / 5 / 5 | 1 / 1 / 1 | 0 / 1 / 1 | 0.00 / 0.94 / 0.94 |
| new-thyroid | 13 / 9 / 9 | 5 / 5 / 5 | 5 / 4 / 4 | 4 / 4 / 4 | 0 / 2 / 2 | 0.00 / 0.93 / 0.93 |
| phoneme | 504 / 341 / 343 | 5 / 5 / 5 | 5 / 5 / 5 | 159 / 99 / 100 | 0 / 98 / 95 | 0.00 / 1.81 / 1.76 |
| pima | 137 / 96 / 87 | 8 / 8 / 8 | 8 / 8 / 8 | 24 / 16 / 14 | 0 / 24 / 30 | 0.00 / 3.12 / 3.91 |
| postoperativ... | 23 / 21 / 20 | 13 / 13 / 12 | 10 / 10 / 9 | 1 / 1 / 1 | 0 / 1 / 1 | 0.00 / 1.39 / 1.39 |
| schizo | 83 / 69 / 66 | 12 / 12 / 12 | 10 / 10 / 10 | 15 / 10 / 10 | 0 / 8 / 9 | 0.00 / 2.35 / 2.65 |
| soybean | 28 / 13 / 13 | 22 / 12 / 12 | 10 / 7 / 7 | 1 / 1 / 1 | 0 / 8 / 8 | 0.00 / 1.29 / 1.29 |
| tae | 41 / 21 / 21 | 5 / 5 / 5 | 5 / 5 / 5 | 13 / 7 / 7 | 0 / 11 / 11 | 0.00 / 10.38 / 10.38 |
| titanic | 336 / 61 / 63 | 8 / 8 / 8 | 8 / 7 / 7 | 61 / 20 / 19 | 157 / 302 / 296 | 7.48 / 14.39 / 14.10 |
| tokyo1 | 46 / 34 / 33 | 24 / 22 / 21 | 16 / 16 / 15 | 10 / 5 / 5 | 0 / 6 / 7 | 0.00 / 0.63 / 0.73 |

*Table 3.* Decision trees used in our experiments: The first entry is for the unpruned tree, the second for the tree computed by the replacement heuristic and the third for the raising heuristic.

