# OpenReview forum: "Optimal Decision Tree Pruning Revisited: Algorithms and Complexity"
_ICML.cc/2025/Conference — ICML 2025 poster_

### Official Review · Reviewer_dtvc · 2025-03-11

**Overall Recommendation:** 3

**Summary:**

The authors investigate the computational complexity of pruning a decision tree.
More specifically the focus is on the algorithmic optimization of two pruning techniques: replacement (removing a subtree and assigning to the root the majority class in the leaves) and raising (removing the subtree rooted at an internal node and substituting it with the subtree rooted at one of its children). The study is on whether there are efficient algorithm that given a maximum number of internal nodes to remove can obtain a pruned tree with at most a given number of misclassification. Parameterized complexity is investigated with respect to single parameters and their combinations.

## Update after rebuttal
I kept my already positive score since there was no additional information from the authors that justified in my opinion an increase.

**Claims And Evidence:**

The mathematical claims are soundly supported by the analyses presented.

**Essential References Not Discussed:**

I have no comments on this

**Experimental Designs Or Analyses:**

The esperiments are meant to provide evidence that the commonly used heuristics are not far from the optimal results given by the DP algorithms. As claimed by the authors it is a small scale study but, nonetheless provide interesting information.

**Methods And Evaluation Criteria:**

The paper presents a theoretical investigation.
The experimental analysis is meant to investigate practically used heuristics with respect to the possible optimization of number of pruning operations and resulting misclassifications allowed. These evaluations is meaningful

**Other Comments Or Suggestions:**

--

**Other Strengths And Weaknesses:**

The paper present a novel perspective on decision tree optimization, from the point of view of pruning.
The hardness results are in my opinion not so surprising.
The DP algorithms are interesting with respect to the idea that in practical cases the number of different thresholds used per feature is small. I think some more should be---maybe experimentally---argued about this point.
However, the complexity attained is significant only with respect to such assumption, since, in practice, could become proportional to nˆ2d.

**Questions For Authors:**

Can you elaborate a bit more on the fact that the parameter D does not explode in practical cases?

**Relation To Broader Scientific Literature:**

The paper appears to take into consideration the relevant papers. In fact, the perspective proposed is, to the best of my knowledge, novel.

**Theoretical Claims:**

I checked the proofs in section 4. Both the DP arguments appear correct.

---

> ### Author Rebuttal · Authors · 2025-04-01
>
> Thank you for your review!
>
> Indeed, we agree that $n^{2d}$ running time would not scale to practical data and this is in particular why we need more precise complexity and running time analyses such as those we provide here. To add to these considerations, note that the implemented algorithm instead of $n^{2d}$ has an exponential running-time part limited by $D_T^{2 d_T} \le D^{2 d_T}$, where $D_T$ and $d_t$ are smaller than $n$ and $d$, respectively, see below:
>
> The maximal number $D$ of thresholds per feature is often much smaller than the number of examples $n$; see Table 3 of Staus et al. (2024, https://arxiv.org/abs/2412.11954) for concrete values. Moreover, often the features are only binary implying that $D=2$. On average, $D$ is smaller than $n/2$.
>
> $D_T$ is the maximal number of different thresholds on cuts in feature $i$ on path $P$ over all features $i \in [d]$ and all root-to-leaf paths $P$. This parameter $D_T$ is much smaller than $n$: In our Table 3 in the appendix we measured $D_T$ for both the unpruned and the pruned trees (there is a typo; the column says $D$ but it should be $D_T$). Together with the values for $n$ from Table 2 we can see that $D_T$ is usually much smaller than $n/10$, in fact in more than half of the instances $D_T< 15$.
>
> Additionally, the exponent in the running time only depends on $d_T$, the maximal number of different features per root-leaf path, which is usually much smaller than $d$; again, see our Tables 2 and 3.
>
> We will emphasize this more in the next version of the paper.

---

### Official Review · Reviewer_x5E5 · 2025-03-11

**Overall Recommendation:** 3

**Summary:**

Decision trees are widely used for tabular datasets. The authors conduct a comprehensive analysis of decision tree pruning operations, including subtree replacement and subtree raising. The paper provides a theoretical analysis of the complexity of each pruning operation, showing that optimal subtree replacement can be achieved in polynomial time and runs linearly in tree size. In contrast, optimal subtree raising is proven to be an NP-complete problem, making it significantly more computationally expensive. The study includes detailed theorems and proofs to support these findings.

**Claims And Evidence:**

Yes.

**Essential References Not Discussed:**

The paper appropriately discusses the related works.

**Experimental Designs Or Analyses:**

Please see above.

**Methods And Evaluation Criteria:**

- Both Table 1 and Figure 10 illustrate the tradeoff between the number of pruned nodes and misclassification error. The results suggest that the heuristic method often appears on the Pareto front constructed by the proposed dynamic programming method, and the conclusion acknowledges that the heuristic is usually sufficient. If the heuristic method already achieves competitive performance with significantly lower computational cost, what is the practical value of using the optimal strategy? The optimal pruning strategy does not seem to provide meaningful performance gains while being more computationally expensive.
- In the optimal tree literature, simple and accurate trees are typically found through search-based methods, and there is not necessarily a tradeoff between simplicity and accuracy. However, in this study, even optimal subtree raising seems to have an almost linear tradeoff between the number of pruned nodes and the misclassification error. Does this suggest an inherent limitation of the subtree raising operation itself? Or does this indicate that the tree induction method is not a good option in the first place?

**Other Comments Or Suggestions:**

NA

**Other Strengths And Weaknesses:**

Please see above.

**Questions For Authors:**

Please see above.

**Relation To Broader Scientific Literature:**

Pruning strategies date back to the origins of tree induction methods (Breiman 1984, Quinlan 1986), where they were introduced to simplify trees and prevent overfitting. This paper presents a comprehensive theoretical analysis of the complexity of different pruning strategies and compares heuristic pruning methods with optimal ones. The paper also relates to the literature on sparse decision trees, which aims to achieve both accuracy and simplicity simultaneously.

**Theoretical Claims:**

The authors provide precise complexity classifications for pruning operations. The claims are supported by detailed proofs.

---

> ### Author Rebuttal · Authors · 2025-04-01
>
> Thank you for your review and the insightful questions!
>
> 1) About the concern about the practical value of using the optimal strategy:
> Prior to our work there was no evidence how good these heuristics work in practice, that is, whether they are close to the optimum, or whether they can be outperformed substantially in quality.
> Our efficient exact algorithms made such an evaluation feasible. Moreover, only with the help of our algorithms such a conclusion that the heuristics are almost always optimal was possible.
> As shown by our hardness results, this is quite surprising and in the future one should investigate this phenomenon further, that is, find explanations as to why these heuristics are so good on these data.
>
> 2) Almost linear tradeoff between the number of pruned nodes and the misclassification error:
> We feel that the intuition of a linear tradeoff might be misleading. Consider Figure 10 in the appendix, where an alternative interpretation is that the relation is rather exponential: Intuitively, pruning one node has a smaller impact in a decision tree of size s than in a tree of size 1. Thus, initially, there is an almost linear dependence on number of raised nodes and classification error, but after a certain barrier the number of errors increases exponentially (as the soybean dataset highlights). This is due the fact that the initial tree is likely overfitting and thus raising a few nodes only leads to few errors but after a certain barrier not enough nodes are left and thus the data cannot be explained anymore, leading to many errors.
> Consequently, we don’t think there is an inherent limitation of these operations. Essentially, one should use them only to eliminate overfitting.

---

### Official Review · Reviewer_mGs2 · 2025-03-13

**Overall Recommendation:** 3

**Summary:**

This manuscript analyzes the complexity gap between two tree pruning strategies: subtree replacement and subtree raising. The former is polynomially solvable, whereas the latter is NP-complete. This paper identifies the key parameters that can bridge the gap between these strategies and analyze their impact, providing precise boundaries between tractable and intractable cases. At last, some numerical experiments are utilized to validate the theorems.

**Claims And Evidence:**

Most of the claims are relatively well-supported but not very clearly presented. Some of the theorems discuss the same concept under different conditions, particularly in Section 5. The authors could consider combining them into a single, more comprehensive theorem to cover all these cases for greater clarity.

Additionally, there is one claim about the Pareto front is not well analyzed. The authors do not specify the multi-objective optimization method used to obtain the Pareto front. Since the pruning problem is an integer programming problem and non-convex, how do the authors ensure that the solutions are (even approximately) Pareto optimal and that the Pareto front is complete? Without a proper analysis, the Pareto front may be inaccurate, leading to potentially incorrect conclusions.

**Essential References Not Discussed:**

No

**Experimental Designs Or Analyses:**

The experimental analysis in this paper is not very strong. If the contribution of this work is to demonstrate that a better solution exists for both error minimization and complexity compared to current methods, the authors should focus more on multi-objective optimization, as this issue stems from non-Pareto optimality.

**Methods And Evaluation Criteria:**

Yes, the pruning strategy is an effective way to improve the performance of decision tree models. The relationship between complexity and accuracy has always been a key concern for researchers. This paper analyzes two pruning strategies and provides valuable insights into reducing complexity.

**Other Comments Or Suggestions:**

1. In Table 1, information about the datasets should be provided (rather than in the appendix), including the number of samples, features, and classes. Although misclassification is used as the objective, it is recommended to report its percentage to give a clearer sense of the error magnitude.
2. The relationship between complexity and accuracy (misclassification errors) can be represented by the Pareto front, which may reveal certain patterns, as demonstrated with the Soybean dataset in Figure 10. However, only a few datasets are analyzed. There may be underlying rules that can be extracted from the Pareto fronts.

**Other Strengths And Weaknesses:**

Strengths:
1. This paper provides interesting insights into the complexity of two pruning strategies, which are useful for improving the performance of decision tree models.
2. Figures 1, 2, and 3 are well-designed and effectively facilitate the understanding of the results.
3. The supplementary materials include historical results.

Weakness:
1. Throughout the paper, the mathematical expressions are not well expressed. This may stem from the problem statement being somewhat difficult to understand and some notations being unclear. Certain notations remain abstract and require further clarification—for example, the domain size $D$.  Additionally, in the first paragraph, the expression $w\in V(T)$ appears. What does $V$ represent? Please ensure that every symbol is clearly defined and well explained.
2. The appendix contains some typos, such as the first line of C.6, "We only show the statement for...”. Please check the appendix.

**Questions For Authors:**

Why do you focus on bi-class problems? What will the results change in multi-class problems?

**Relation To Broader Scientific Literature:**

This paper contributes to the complexity analysis of pruning strategies, specifically, subtree replacement and subtree raising. These strategies enhance the performance of decision tree models and can be applied to various heuristic approaches such as CART and C4.5.

**Theoretical Claims:**

The authors provide extensive analysis to illustrate and prove the theorems. However, some theorems and lemmas could be combined—for instance, Lemma 4.4, Theorem 4.5, and Theorem 4.6, as they all discuss the time complexity of $DTR_{AIS_{\geq}}$under different conditions. It is recommended to consolidate them into a single theorem for clarity and conciseness. A similar issue exists in Section 5, which analyzes whether $DTR_{AIS_=}$ and $DTR_{AIS_{\geq}}$ are solvable (time complexity) under different critical parameter conditions and the ETH assumption.

---

> ### Author Rebuttal · Authors · 2025-04-01
>
> Thank you for your feedback! Allow us to respond to all your concerns:
>
>
> Presentation:
>
> - Theorems in Section 4 and 5: Note that all of these theorems correspond either to different combinations of parameters or different subsets of parameters that are fixed to at most some constant value. We have invested quite a lot of time into how to present them in an accessible way; Figures 2 and 3 are highly condensed versions of the statements that still convey most of the information. We feel that merging the statements into one would indeed decrease the readability since it would be more difficult to match the proofs to the corresponding statements.
>
> - The $V$ in $w \in V(T)$ is a common notation for the vertex set of the tree $T$; we have clarified this.
>
> - The domain size $D$ is the maximum number of different values that a feature can attain, we have also clarified this.
> We have already incorporated your specific feedback into our local version of the submission and will ensure that the presentation of the final version is as accessible as possible.
>
>
> Pareto-front:
>
> - We indeed correctly compute the Pareto front or, more precisely, for each $k$ the point $(t, k)$ such that it is possible to prune exactly $k$ nodes to achieve exactly t errors and every solution with at most $k$ pruned nodes has at least $t$ errors: Any algorithm solving the search problems DTRAIS=> and DTRAIS= can also be applied to finding the optimum values for the corresponding optimization problems (see also the introduction). That is, with such an algorithm we can minimize the number of errors for fixed number $k$ of pruning operations simply by incrementing the error parameter $t$, starting with 0, until the algorithm returns a solution. Solving these optimization problems for all $k$ will yield the Pareto front. Since our algorithms provably solve the search problems correctly, we are certain to find the optimum values. We will emphasize this more clearly in the final version.
>
>
> Datasets:
>
> - Indeed, it would be desirable to extend the experiments; however, this is out of scope for the current paper where we mainly want to develop the fundamental theory. Thus we leave this for future work.
>
> - We agree, it would be good to add more info from the appendix to the overview of the datasets given in Section 6. If the paper were to be accepted, we would be happy to utilize the extra page for this purpose; in any case, we will make a full version available on arxiv or another public repository.
>
>
> Biclass vs. multiclass:
>
> - The focus on the biclass setting is for clarity of the theory. All our hardness results directly apply to the multiclass setting. All our dynamic-programming algorithms and the algorithms for Lemma 4.4 and Theorem 4.5 also directly apply to the multiclass setting without changes. The algorithm for Theorem 4.6. can easily be adapted to the multiclass setting without changing the running time: In the tree-enumeration algorithm, when introducing a new leaf, we set the leaf to the class of the dirty example. The remaining algorithm remains the same. The algorithm in Theorem 3.1 can straightforwardly be adapted to the multiclass setting: In the first step we instead compute, for each node in the tree and each class $c$, the number of misclassified examples in this node, if the node were to be replaced by a leaf with class $c$. The rest of the algorithm remains the same.

---

### Official Review · Reviewer_iXCd · 2025-03-14

**Overall Recommendation:** 4

**Summary:**

The submission at hand aims at understanding the parameterized complexity of problems relating to editing decision trees to conform to a data set up to a bounded number of errors. The considered operations are either of raising and replacing and the considered parameters are numbers given on input, relate to the size of the input or relate to the number of features with certain properties. Apart from few specific combination the obtained parameterized complexity theoretic classification is exhaustive with respect to the combination of all considered parameters and utilizes a number of maybe not particularly groundbreaking but diverse and well applied algorithmic techniques and hardness reductions.
The theoretical results are complemented by a small empirical section on the quality of common heuristics for minimizing a number of edits given a certain error budget. These experiments might weakly indicate some weaknesses of the considered heuristics that deserve further investigation.

**Claims And Evidence:**

No concerns.

**Essential References Not Discussed:**

None that I am aware of.

**Experimental Designs Or Analyses:**

I only read what is not in the appendix.

**Methods And Evaluation Criteria:**

Appropriate.

**Other Comments Or Suggestions:**

- Maybe specify that you assume decision trees to be binary
- Line 417(right): replace both operations by each operation

**Other Strengths And Weaknesses:**

Overall, this submission is topically relevant and solid in terms of its contributions and presentation. I recommend acceptance.

**Questions For Authors:**

Which of these results can be adapted when allowing both replacing and raising?

**Relation To Broader Scientific Literature:**

This submission studies the complexity theoretic behavior of a problem relevant to ML. There have not been extremely many papers like this on this specific kind of problem but there is much precedent of similar analyses for a range of problems.

**Theoretical Claims:**

I had at least a superficial look at all proofs and am convinced by their correctness up to minor details.

---

> ### Author Rebuttal · Authors · 2025-04-01
>
> Thank you for the review and the helpful feedback!
>
> We now make clear in the introduction that we focus on binary trees and incorporated the comment about line 417.
>
>
>
> > Which of these results can be adapted when allowing both replacing and raising?
>
>
> That’s an insightful question. While the details depend slightly on how the problems are formulated (e.g., do we have a budget for each operation on how many cuts should be pruned or do we only care about the total number), our algorithmic and hardness results for the raising operation are generally adaptable to support also the replacement operation.
> As to hardness, observe that in our reduction for showing the hardness of the raising problem (Theorem 5.1), each subtree rooted at an inner node has both a blue and a red leaf. Thus, a replacement operation on an inner node that replaces it by a red (correspondingly, blue) leaf can be simulated by a sequence of raising operations that raises a red (blue) leaf of the subtree in the place of the inner node. Consequently, allowing also replacement operations does not make the problem any easier or harder.
> For our algorithms, we would need to adapt the dynamic programming recurrences to take into account the possibility of replacing the whole subtree, but other than that the details are similar. More precisely, recurrence (1) of the proof of Theorem 1 now has a fourth case to consider a tree replacement operation.
> In summary, allowing both replacement and raising would yield hardness and tractability results similarly to when allowing only raising operations.

---

### Decision · Program_Chairs · 2025-05-01

**Decision:**

Accept (poster)

**Comment:**

All reviewers suggested acceptance so I will concur with them. That said, I think the results are technically interesting but of narrow impact. The fundamental reason for a-posteriori pruning a CART-style decision tree is that CART doesn't optimize a loss function over the tree, and one gets better trees (with lower generalization error) by growing them very deep (so they overfit) and then pruning them. If pruning is defined as subtree replacement, the traditional and widely used algorithm (CART cost-complexity pruning) solves this exactly and efficiently. I think it is unlikely that other forms of pruning (with more difficult pruning operations, such as subtree raising) will improve much the performance in practice, and the small experiments in the paper don't clarify this. It would be useful to readers if the authors comment on the practical impact of their results in their "Outlook" section.

A recent line of work not discussed in the paper is based on Tree Alternating Optimization (TAO). This makes it possible to optimize a loss function plus sparsity regularization (converging to a local optimum) and it scales to large sample sizes and feature dimensionality and large trees (unlike dynamic programming or MIO approaches). Pruning happens during training automatically as a result of the sparsity regularization.